# Position-Query-Based Autoencoders for View Decoupled Cross Point Cloud Reconstruction and a Self-Supervised Learning Framework

## Abstract

Point cloud learning, especially in a self-supervised way without manual labels, has received emerging attention in both vision and learning communities, with its potential utility in wide areas. Most existing generative approaches for point cloud self-supervised learning focus on recovering masked points from visible ones within a single view. Recognizing that a two-view pre-training paradigm inherently introduces greater diversity and variance, it could thus enable more challenging and informative pre-training. Inspired by this, we explore the potential of two-view learning in this domain. In this paper, we propose **Point-PQAE**, a cross-reconstruction generative paradigm that first generates two decoupled point clouds/views and then reconstructs one from the other. To achieve this goal, we develop a crop mechanism for point cloud view generation for the first time and further propose a novel positional encoding to represent the 3D relative position between the two decoupled views. The cross-reconstruction significantly increases the difficulty of pre-training compared to self-reconstruction, which enables our method to achieve new state-of-the-art results and surpasses previous single-modal self-reconstruction methods in 3D self-supervised learning by a margin. Specifically, it outperforms self-reconstruction baseline (Point-MAE) 6.5%, 7.0%, 6.7% in three variants of ScanObjectNN with Mlp-Linear evaluation protocol. Source code will be released.

## 1 Introduction

3D vision is gaining increasing attention for its wide applications such as autonomous driving (Qian et al., 2022b) and robotics (Sergiyenko & Tyrsa, 2020; Enebuse et al., 2021), owing to its ability to understand the human world. The point cloud is the most popular representation form of data in 3D vision and many analyses based on point cloud are explored to solve various tasks such as object classification (Qi et al., 2017a;b; Wang et al., 2019; Li et al., 2018b; Guo et al., 2021), object detection (Misra et al., 2021; Qi et al., 2019; Zhou & Tuzel, 2018) and segmentation (Qi et al., 2017a; Wang et al., 2019; Li et al., 2018b; Guo et al., 2021; Wu et al., 2023), they typically necessitate fully-supervised training from scratch. Compared to 2D data, point cloud generally requires more expensive and labor-intensive efforts to collect and annotate, which refrains the development of the fully supervised 3D representation learning method. Self-supervised learning is one of the predominant approaches to address this issue and is proven to be effective in 2D vision (Chen et al., 2020; He et al., 2022; Caron et al., 2021; Zhou et al., 2021; He et al., 2020; Grill et al., 2020). Inspired by this, self-supervised learning has been widely studied in the 3D field in recent years.

Self-supervised learning in 3D can be mainly divided into two categories, *i.e.*, generative methods (Chen et al., 2024; Pang et al., 2022; Wang et al., 2021a; Li et al., 2018a; Achlioptas et al., 2018; Zhang et al., 2022; Liu et al., 2022) and contrastive methods (Afham et al., 2022; Xie et al., 2020; Dong et al., 2022; Qi et al., 2023; Huang et al., 2021; Sanghi, 2020; Du et al., 2021). Similar to 2D, **contrastive methods** in 3D aim to learn global-discriminative information by maximizing the mutual information across views (Afham et al., 2022; Xie et al., 2020; Dong et al., 2022) with intra-/inter- modal information. PointContrast (Xie et al., 2020) is the first to learn transformation invariance across views through the extension of InfoNCE objective (Oord et al., 2018; Xie et al., 2020). Different from contrastive methods, The core idea of the **generative methods** (Pang et al.,

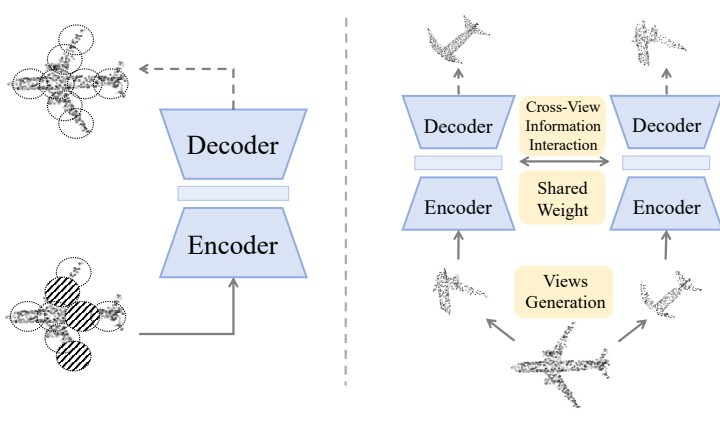

Self-Reconstruction        Cross-Reconstruction (Ours)

Figure 1: Comparison between self-reconstruction (Point-MAE (Pang et al., 2022) and Point-MAE-based methods) and cross-reconstruction (Ours) paradigms. In the self-reconstruction paradigm, part of the input data is masked, and the autoencoder is trained to recover the missing patches. In contrast, the cross-reconstruction paradigm generates decoupled views first, with one view leveraging cross-view information to reconstruct the other, using an autoencoder backbone.

2022; Zhang et al., 2022; Liu et al., 2022) in 3D is inspired by MAE (He et al., 2022), which masks the input point cloud and uses the remaining visible part to reconstruct the masked part. Point-MAE (Pang et al., 2022) is one of the conventional methods, which learns point-wise information through the self-reconstruction mechanism. Specifically, Point-MAE masks a high ratio of the point cloud and reconstructs the masked points through the visible part. On the basis of Point-MAE, Point-M2AE (Zhang et al., 2022) proposes a hierarchical Transformer (Liu et al., 2021) with a corresponding masking strategy, capturing both fine-grained and high-level semantics of 3D shapes.

Existing generative methods mostly focus on the masked reconstruction in a single view, where single-view learning has been proven less difficult and informative than two views (Chen et al., 2020; Grill et al., 2020) since two views would bring more variance than the single case. To bring the benefits of the two-view learning paradigm to the generative learning field, one straightforward assumption is adopting the two-view learning paradigm in the 3D generative pertaining tasks to increase the difficulty of point-wise reconstruction. However, unlike 2D vision data, constructing two views in the 3D point cloud is more challenging, as point cloud data is less structured. Besides, cross-reconstruction (reconstructing one view from the other view) is much more difficult than self-reconstruction (reconstructing the masked single view), which involves learning both the inter-positional relation between two views and the inner spatial information within the view. The differences between these two paradigms can be seen in Figure 1. Aimed at solving the above issues, we propose Point-PQAE (**P**osition **Q**uery based **A**uto**E**ncoder for **Point** Cloud), a novel framework aimed at addressing two primary challenges: i) how to construct two views in point cloud data, and ii) how to perform cross-reconstruction between two views. As shown in Figure 2, for each point cloud, we first apply a custom-designed point cloud crop mechanism, which randomly selects two points as the central points of two views and records their positions. Following this, for each central point, the nearest points according to a specified ratio are incorporated into the view. Subsequently, we normalize the cropped point cloud and apply random augmentation (*e.g.*, rotation) to each point cloud to achieve view decoupling. Then we calculate the relative positional embedding (RPE) for these two decoupled views. Finally, we take the RPE and one view of the point cloud as input to predict the point-wise input of the other view. **The highlights of this paper are:**

i) **New generative framework:** We propose Point-PQAE, the first framework that successfully brings cross-reconstruction to 3D generative field for point cloud self-supervised learning, with three modules: 1) decoupled views generation, 2) RPE generation, and 3) positional query block. To our knowledge, we are the first to design and apply crop mechanism to point cloud self-supervised learning. Our framework, Point-PQAE, enables more informative and challenging self-supervised pre-training compared to existing traditional self-reconstruction methods.

ii) **The first relative position-aware query module for point cloud:** We introduce a positional query block after the encoder to capture relative position information between decoupled views.

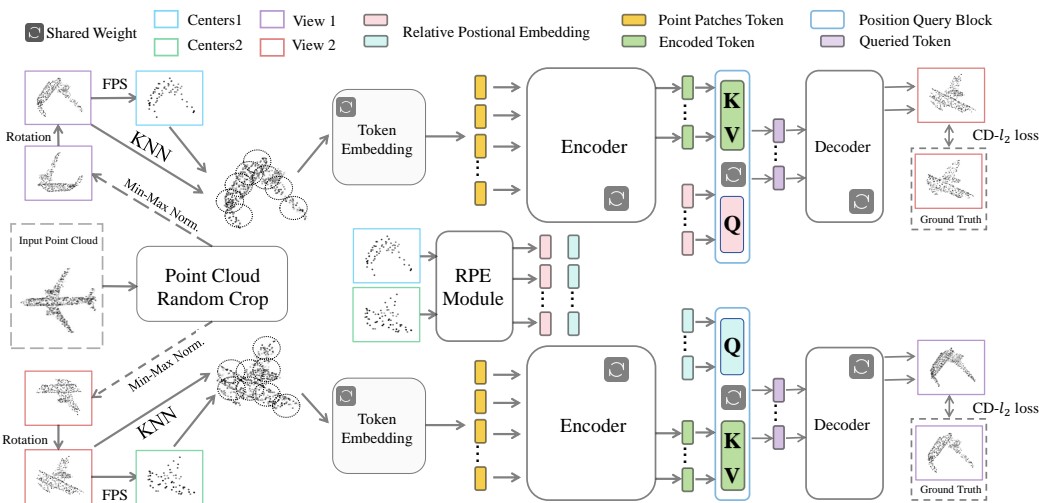

Figure 2: Pipeline of Point-PQAE. The input point cloud is randomly cropped followed by the rotation to generate new views. Then, we feed the views to the patch embedding layer and the transformer encoder, followed by the proposed positional query block. The Relative Positional Embedding (RPE) is obtained through the RPE module by extracting relative geometric relations, which is taken as "Query" in the cross-attention mechanism. The queried hidden embeddings are then simply fed to the decoder to predict the inputs of the other view.

The module applies cross-attention between the hidden representations of one view and the relative positional embeddings, enabling it to predict the other view. Our module can be easily plugged into various related tasks, $e.g.$, distillation as stated in Appendix D.

iii) **Strong performance:** Empirically, Point-PQAE achieves new state-of-the-art performance on several benchmarks, $e.g.$, outperforming all published methods on few-shot learning and achieving average improvements of 1.8%, 6.7%, and 4.4% over the baseline (Point-MAE) on ScanObjectNN classification across three evaluation protocols (FULL, MLP-LINEAR, MLP-3), respectively.

## 2 RELATED WORK

We briefly review the literature on self-supervised learning in vision and point cloud.

**SSL in 2D Vision.** Self-supervised learning has been well-explored in 2D vision. These methods can be roughly divided into two categories, $i.e.$, Masked-Image-Modeling-based (MIM-based) and Contrastive-based. *Contrastive methods* usually learn discriminative information by crafting two different views through random augmentation and learning the invariance of two views. Sim-CLR (Chen et al., 2020) is one of the conventional methods, which takes the views augmented from the same image as the positive pair, and the views augmented from different images in the same mini-batch as the negative pairs. MoCo (He et al., 2020) utilizes a memory bank to store these negative samples and further proposes a momentum-updated mechanism to avoid model collapse. To further reduce the effect of negative pairs, BYOL (Grill et al., 2020) borrows the idea in reinforcement learning, using online and target networks and Exponential Moving Average (EMA) to update the target network. BYOL directly adds the $l_2$ objective on the target output and online output, avoiding the use of negative pairs. Depart from contrastive learning, *MIM-based methods* usually mask the input image and make models to reconstruct the masked area, which could enable models to capture fine-grained information. MAE (He et al., 2022) and SimMIM (Xie et al., 2022) are the two widely-known methods belonging to the MIM-based family. MAE divides the network into encoder and decoder parts. Then, MAE masks the image and directly drops this masked area, followed by the encoder-decoder to reconstruct. Different from MAE, SimMIM (Xie et al., 2022) replaces these masked patches with a learnable parameter to reconstruct these masked patches. Following MAE and SimMIM, variant directions are well explored, $e.g.$, masking strategy (Zhang et al., 2023c; Li et al., 2022) and reconstruct feature map (Zhou et al., 2021).

**Representation Learning for Point Clouds.** It is a challenging task due to its irregular and sparse nature when compared to other modalities such as images which are well-structured and uniform.

PointContrast (Xie et al., 2020) firstly borrows the idea of contrastive learning in 2D and performs point-level invariant mapping on two transformed views of the given point cloud. On the basis of PointContrast, CrossPoint (Afham et al., 2022) further proposes inter and intra-modal contrastive objectives to enhance the global-discriminative capability. To better model multi-modal data, ACT (Dong et al., 2022) employs cross-modal auto-encoders as teacher models to acquire knowledge from other modalities. Depart from contrastive objectives, Point-MAE (Pang et al., 2022) proposes to randomly mask points, and reconstruct the masked part through the visible part. Following Point-MAE and CrossPoint, ReCon (Qi et al., 2023) harmoniously incorporates the generative method Point-MAE and the contrastive method. **Almost all the existing generative methods in 3D rely on self-reconstruction.** To this end, we propose a new cross-reconstruction generative framework, named Point-PQAE, which increases the difficulty of pre-training by introducing reconstruction between two decoupled views into training, bringing more variance to the training samples, as opposed to within a single view. This requires the model to effectively learn both intra-view and inter-view knowledge, thereby enabling the FPS model to learn more semantic representations.

# 3 THE PROPOSED POINT-PQAE

## 3.1 POINT CLOUDS VIEWS AND EMBEDDINGS GENERATION

**Decoupled Views Generation.** 2D image Random Crop has been widely used in both the supervised domain (He et al., 2016; Simonyan & Zisserman, 2014; Krizhevsky et al., 2012) and the self-supervised domain (Chen et al., 2020; He et al., 2020; Bao et al., 2021), and it has been proved to be effective in practice. This method typically involves randomly selecting a rectangular region from an image and cropping it. However, to the best of our knowledge, there is no corresponding crop mechanism proposed in 3D, especially for self-supervised representation learning. Instead of directly bringing the crop algorithm in 2D to 3D (directly choose a random cube in space), which will bring inconsistent problems (the points contained by the cubes of the same size may change intensively due to the variety of point cloud distribution in 3D), we design in a smarter way: For a randomly selected crop ratio, we first select a center point randomly and then choose the corresponding number of points in the point cloud that are closest to the center point.

Formally, given an input point cloud with p points $\mathbf{X} \in \mathbb{R}^{p \times 3}$ and a pre-defined minimum crop ratio $r_m$ where $0 < r_m < 1$, we first randomly select the crop ratios $r_1$ and $r_2$ uniformly from $[r_m, 1]$. Then, we randomly sample two center points $\mathbf{C_1} \in \mathbb{R}^{1 \times 3}$ and $\mathbf{C_2} \in \mathbb{R}^{1 \times 3}$ and get $\mathbf{X}_1$ and $\mathbf{X}_2$ by expanding nearest $r_1 \times p$ and $r_2 \times p$ points to the center point $\mathbf{C_1}$ and $\mathbf{C_2}$ respectively in $\mathbf{X}$. Meanwhile, we record the absolute coordinates of cropped point clouds' geometric centers, which are denoted as $\mathbf{L}_1 \in \mathbb{R}^{1 \times 3}$ and $\mathbf{L}_2 \in \mathbb{R}^{1 \times 3}$, respectively. The process can be formulated as:

$$r_1, r_2 = \text{UniformSample}([r_m, 1]), \quad \mathbf{C_1}, \mathbf{C_2} = \text{RandomSelect}(\mathbf{X}), \tag{1}$$

$$\mathbf{X}_1 = \text{N-Nearest}(\mathbf{C_1}, r_1, \mathbf{X}), \quad \mathbf{X}_2 = \text{N-Nearest}(\mathbf{C_2}, r_2, \mathbf{X}) \tag{2}$$

where $\mathbf{X}_1 \in \mathbb{R}^{r_1 p \times 3}$ and $\mathbf{X}_2 \in \mathbb{R}^{r_2 p \times 3}$ mean the two cropped views. The $\mathbf{X}_1$ and $\mathbf{X}_2$ are normalized using the min-max normalization, centered on $\mathbf{L_1}$ and $\mathbf{L_2}$, respectively. Subsequently, we apply additional augmentation to further enhance the variance between two views. Specifically, we apply random rotations to the cropped views. The distinct normalization centers, along with random rotation, effectively isolate the coordinate systems of the two views and alter their fixed relative relationships. This process decouples the two point clouds, resulting in two independent views.

**Point Patches Generation.** Following Point-BERT (Yu et al., 2022), we divide the input point cloud into irregular point patches (may overlap) via the Farthest Point Sampling(FPS) and K-Nearest Neighborhood (KNN) algorithm. Formally, given two point clouds (views) $\mathbf{X}_1$ and $\mathbf{X}_2$, we first use the FPS algorithm to sample pre-defined $n$ points as group centers, which we called $\mathbf{G}_1 \in \mathbb{R}^{n \times 3}$ and $\mathbf{G}_2 \in \mathbb{R}^{n \times 3}$, respectively. Then, based on the center points, we use KNN to choose the predefined $k$-nearest points for each group center $\mathbf{G}^{(i)}$ $(0 \leq i \leq n)$ and finally get $n$ groups (each group contains $k$ points). We apply the FPS and KNN on $\mathbf{X}_1$ and $\mathbf{X}_2$, respectively:

$$\mathbf{G}_1 = \text{FPS}(\mathbf{X}_1), \quad \mathbf{G}_2 = \text{FPS}(\mathbf{X}_2) \tag{3}$$

$$\mathbf{P}_1 = \text{KNN}(\mathbf{X}_1, \mathbf{G}_1), \quad \mathbf{P}_2 = \text{KNN}(\mathbf{X}_2, \mathbf{G}_2) \tag{4}$$

where $\mathbf{G}_1, \mathbf{G}_2 \in \mathbb{R}^{n \times 3}$ and $\mathbf{P}_1, \mathbf{P}_2 \in \mathbb{R}^{n \times (k \cdot 3)}$. Then, the $\mathbf{P}_1$ and $\mathbf{P}_2$ both are treated as a sequence of length $n$ and dimension $k \cdot 3$ as the input.

## 3.2 BACKBONE WITH POSITIONAL QUERY

Following (Pang et al., 2022), we adopt a lightweight PointNet (Qi et al., 2017a), followed by the standard Transformer with an asymmetric encoder-decoder structure in the Point-PQAE. After the encoder projects the input embedding into the latent space, we use the latent representation of one view and relative positional embedding (RPE) between two views to implement the position query through cross-attention (Vaswani et al., 2017). We then directly feed the output of the position query block to the decoder to reconstruct another view without using extra positional embedding. The last layer of the autoencoder adopts a simple prediction head to achieve the cross-reconstruction target.

**Encoding.** The encoder of Point-PQAE consists of the lightweight PointNet and a standard Transformer blocks (Vaswani et al., 2017), which projects $\mathbf{P}_1$ and $\mathbf{P}_2$ to latent space, respectively:

$$\mathbf{H}_1 = \text{Encoder}(\mathbf{P}_1, \mathbf{G}_1), \quad \mathbf{H}_2 = \text{Encoder}(\mathbf{P}_2, \mathbf{G}_2), \tag{5}$$

where $\mathbf{H}_1 \in \mathbb{R}^{n \times D}$ and $\mathbf{H}_2 \in \mathbb{R}^{n \times D}$ mean the latent representations of the view 1 and view 2. Note that $D$ is the hidden dimension of the networks.

**Positional Query Block.** After obtaining latent representations $\mathbf{H}_1$ and $\mathbf{H}_2$, we feed them into the proposed positional query block to extract cross-information. The module can be mainly divided into 3 sub-stages: attain relative information, RPE generation, and cross-attention.

*Attain Relative Information.* Take reconstructing view 2 from view 1 as an example. To reconstruct view 2, we have to know the latent representations of view 1 and the relative positions of view 2 centered on view 1, where the latent representations of view 1 can be obtained through the encoder network. Therefore, the key to reconstructing view 2 is to obtain the relative positions of view 2 centered on view 1. Back to the crop operation, we have recorded the geometric centers of views, denoted as $\mathbf{L}_1$ and $\mathbf{L}_2$, respectively. We define the relative position of view 2 centered on view 1 as $\mathbf{RL}_{1 \to 2} = \mathbf{L}_1 - \mathbf{L}_2$ where $\mathbf{RL}_{1 \to 2} \in \mathbb{R}^{1 \times 3}$. To better align the dimension in the latter operation, we expand the $\mathbf{RL}_{1 \to 2}$ to $\mathbb{R}^{n \times 3}$ by repeating the content for $n$ times. By obtaining the relative positions between the two views, we further add the group center location to help the PQ modules reconstruct each group of the inputs. Specifically, we define the group-/patch- wise relative positions as:

$$\mathbf{RP}_{1 \to 2} = \text{ConCat}(\mathbf{G}_2, \mathbf{RL}_{1 \to 2}), \quad \mathbf{RP}_{2 \to 1} = \text{ConCat}(\mathbf{G}_1, \mathbf{RL}_{2 \to 1}) \tag{6}$$

where $\mathbf{RP} \in \mathbb{R}^{n \times 6}$ (as $\mathbf{G} \in \mathbb{R}^{n \times 3}$ and $\mathbf{RL} \in \mathbb{R}^{n \times 3}$) and $\text{ConCat}(\cdot, \cdot)$ is the concatenate operation.

*Relative Positional Embedding (RPE).* Instead of simply using learnable positional encoding, which could hurt the expression of the relative positions between the two views (see ablation in Sec. 4.3), we use the fixed positional encoding of relative information to cut the uncertainty. Specifically, for elements in the second dimension of $\mathbf{RP}_{1 \to 2}$, we use a popular form (He et al., 2022) in Transformers to generate relative positional embedding $\mathbf{RPE}_{1 \to 2} \in \mathbb{R}^{n \times D}$:

$$\mathbf{RPE}_{1 \to 2}^i = \left[ \sin\left( \frac{\mathbf{RP}_{1 \to 2}^i}{e^{2 \times \frac{1}{D/12}}} \right), \cos\left( \frac{\mathbf{RP}_{1 \to 2}^i}{e^{2 \times \frac{1}{D/12}}} \right), \sin\left( \frac{\mathbf{RP}_{1 \to 2}^i}{e^{2 \times \frac{2}{D/12}}} \right), \right.$$
$$\left. \cos\left( \frac{\mathbf{RP}_{1 \to 2}^i}{e^{2 \times \frac{2}{D/12}}} \right), \ldots, \sin\left( \frac{\mathbf{RP}_{1 \to 2}^i}{e} \right), \cos\left( \frac{\mathbf{RP}_{1 \to 2}^i}{e} \right) \right] \tag{7}$$

where $\mathbf{RP}_{1 \to 2}^i \in \mathbb{R}^{n \times 1}$, $\mathbf{RPE}_{1 \to 2}^i \in \mathbb{R}^{n \times (D/6)}$ ($1 \leq i \leq 6$). $e = 10000$ is the pre-defined parameter, which is also used in MAE (He et al., 2022). Finally, we concatenate $\mathbf{RPE}_{1 \to 2}^i$ for each dimension to obtain $\mathbf{RPE}_{1 \to 2} \in \mathbb{R}^{n \times D}$.

*Cross-Attention.* Empirically, assume the latent representations of view 1, denoted as $\mathbf{H}_1$, contains intrinsic features and the global characteristics of its source point cloud, then with the combination of the relative positional embeddings $\mathbf{RPE}_{1 \to 2}$, the view 2 can be reconstructed. Specifically, we employ $\mathbf{RPE}_{1 \to 2}$ (as $\mathbf{Q}$) and $\mathbf{H}_1$ (as $\mathbf{K}, \mathbf{V}$) in cross-attention mechanisms, as can be formalized:

$$\mathbf{T}_2 = \text{Attn}(\mathbf{Q}_{\mathbf{RPE}_{1 \to 2}}, \mathbf{K}_{\mathbf{H}_1}, \mathbf{V}_{\mathbf{H}_1}) = \text{Softmax}\left( \frac{\mathbf{Q}_{\mathbf{RPE}_{1 \to 2}} \mathbf{K}_{\mathbf{H}_1}}{\sqrt{D}} \right) \mathbf{V}_{\mathbf{H}_1} \tag{8}$$

where $\mathbf{Q}_{\mathbf{RPE}_{1 \to 2}} = \mathbf{RPE}_{1 \to 2} \mathbf{W}_{\mathbf{Q}}$, $\mathbf{K}_{\mathbf{H}_1} = \mathbf{H}_1 \mathbf{W}_{\mathbf{K}}$, and $\mathbf{V}_{\mathbf{H}_1} = \mathbf{H}_1 \mathbf{W}_{\mathbf{V}}$, where $\mathbf{W}_{\mathbf{Q}}$, $\mathbf{W}_{\mathbf{K}}$, and $\mathbf{W}_{\mathbf{V}}$ are learnable parameters. Reconstructing view 1 from view 2 is a siamese condition of this, through the same way as stated above, we can obtain $\mathbf{RL}_{2 \to 1}$, $\mathbf{RP}_{2 \to 1}$, $\mathbf{RPE}_{2 \to 1}$ and yield $\mathbf{T}_1$.

**Decoding.** The obtained $\mathbf{T}_1$ and $\mathbf{T}_2$ are fed into a decoder composed of a few transformer blocks:

$$\mathbf{Z}_1 = \text{Decoder}(\mathbf{T}_1), \quad \mathbf{Z}_2 = \text{Decoder}(\mathbf{T}_2) \tag{9}$$

Finally, similar to (Pang et al., 2022), we adopt a projection head (composed of a fully connected layer) to predict the input point cloud as follows, where $\mathbf{P}_{pred}^1 \in \mathbb{R}^{n \times k \times 3}$ and $\mathbf{P}_{pred}^2 \in \mathbb{R}^{n \times k \times 3}$:

$$\mathbf{P}_{pred}^1 = \text{Reshape}(\text{Linear}(\mathbf{Z}_1)), \quad \mathbf{P}_{pred}^2 = \text{Reshape}(\text{Linear}(\mathbf{Z}_2)) \tag{10}$$

## 3.3 OBJECTIVE FUNCTION

The overall objective is to perform cross-reconstruction, which involves predicting view 1 from view 2 and predicting view 2 from view 1. Given the predicted point patches $\mathbf{P}_{pred}^1$ and $\mathbf{P}_{pred}^2$ and ground truth $\mathbf{P}_1$ and $\mathbf{P}_2$, we compute the cross-reconstruction loss using the $l_2$ Chamfer Distance loss function (Fan et al., 2017), written as $\mathcal{L}_{cross} = \mathcal{L}_{2 \rightarrow 1} + \mathcal{L}_{1 \rightarrow 2}$, where

$$\mathcal{L}_{2 \rightarrow 1} = \frac{1}{\left|\mathbf{P}_{pred}^1\right|} \sum_{a \in \mathbf{P}_{pred}^1} \min_{b \in \mathbf{P}_1} \|a - b\|_2^2 + \frac{1}{|\mathbf{P}_1|} \sum_{b \in \mathbf{P}_1} \min_{a \in \mathbf{P}_{pred}^1} \|a - b\|_2^2 \tag{11}$$

where $|\mathbf{P}|$ is the cardinality of the set $\mathbf{P}$ and $\|\cdot\|_2$ is the $l_2$ distance. The $\mathcal{L}_{1 \rightarrow 2}$ follows similarly.

## 4 EXPERIMENTS

This section is organized as follows: First, we present the model architecture and pre-training details. Second, we validate the effectiveness of our pre-trained model on a wide range of downstream tasks, including object classification, few-shot learning, and part segmentation. Finally, ablation studies are conducted to demonstrate the properties and robustness of our proposed Point-PQAE.

### 4.1 PRE-TRAINING SETUPS

**Pre-trained Dataset.** We use dataset ShapeNet (Chang et al., 2015) for pre-training of Point-PQAE following previous studies (Pang et al., 2022; Dong et al., 2022; Qi et al., 2023). ShapeNet (Chang et al., 2015) consists of about 51,300 clean 3D models, covering 55 common object categories.

**Model Structure.** We adopt standard Transformer blocks (Vaswani et al., 2017) in the autoencoder's backbone where the encoder has 12 Transformer blocks and the decoder has 4 blocks. MLP ratio in Transformer blocks is set to 4. Each Transformer block has 384 hidden dimensions and 6 heads. The positional query module between encoder and decoder only performs cross-attention mechanism once as stated in Section 3.2, and the number of heads is set to 6 and remains 384 hidden dimensions.

**Experiment Detail.** For each input point cloud, we only apply random rotation for pre-training data augmentation. After sampling 1024 points via FPS from the input point cloud, we generate two different partial point clouds by randomly cropping and rotating as illustrated in Section 3.1 with $r_m = 0.6$. Both of them are divided into 64 patches with 32 points via FPS and KNN. The Point-PQAE model undergoes pre-training for 300 epochs using an AdamW (Loshchilov & Hutter, 2017) optimizer with a batch size of 128. The initial learning rate is set to 0.0005, with a weight decay of 0.05, with cosine learning rate decay (Loshchilov & Hutter, 2016) utilized.

We visualize the pre-training results in Figure 3. It shows Point-PQAE learns cross-knowledge well and is able to generalize excellently to other crop ratios though pre-trained with $r_m = 0.6$.

### 4.2 TRANSFER LEARNING ON DOWNSTREAM TASKS

**Transfer Protocol.** We use three transfer learning protocols for classification tasks following (Dong et al., 2022): FULL, MLP-LINEAR, MLP-3, which are detailed inAppendix C.

**3D Real-World Object Classification.** Models' transfer ability can be well demonstrated through testing on a 3D real-world object dataset. Therefore, we transfer our pre-trained model to ScanObjectNN (Uy et al., 2019) for classification, which is one of the most challenging 3D datasets covering approximately 15,000 real-world objects across 15 categories. We conduct experiments on three variants: OBJ-BG, OBJ-ONLY, and PB-T50-RS. The results are reported in Table 1. Our Point-PQAE surpasses previous single-modal self-supervised methods by a margin under all protocols. In contrast to the most relevant baseline Point-MAE, Point-PQAE significantly outperforms it, showing

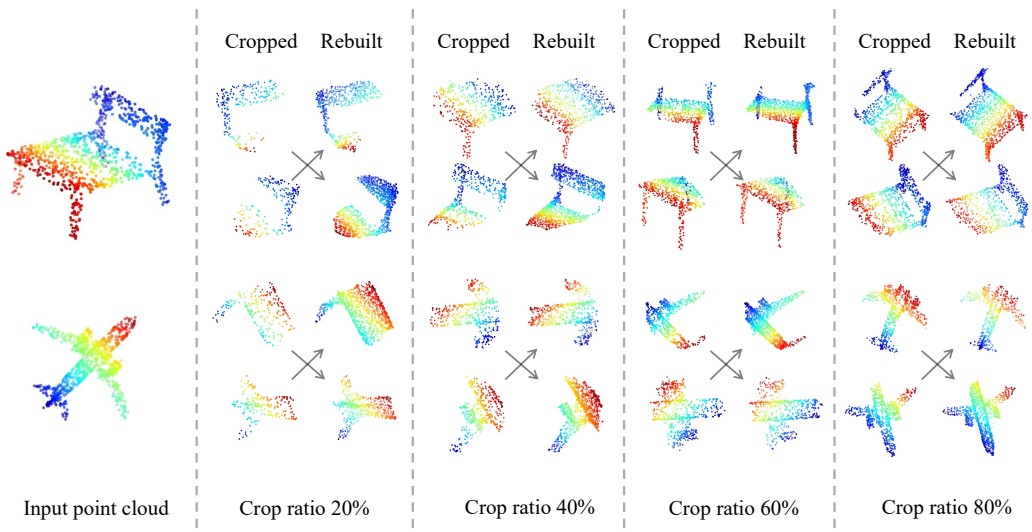

Figure 3: Cross-reconstruction results on ShapeNet. The arrow points from source point clouds to cross-reconstruction results. Point-PQAE generalizes well to other crop ratios though with minimum crop ratio $r_m = 0.6$ when pre-training.

improvements of **2.4%**, **1.7%**, and **1.2%**. Especially for the MLP-LINEAR and MLP-3 protocols, our Point-PQAE outperforms Point-MAE by **6.7%** and **4.4%** on average, respectively. The great frozen representation evaluation results indicate the superior quality of the representation learned by our Point-PQAE during pre-training. It shows that the cross-reconstruction paradigm proposed by us learns more robust representations and generalizes better on real-world datasets.

**3D clean Object Classification.** We evaluate the performance of our pre-trained model for object classification using the ModelNet40 dataset (Wu et al., 2015). ModelNet40 comprises 12,311 meticulously crafted 3D CAD models, representing 40 distinct object categories. During testing, we adhere to the standard voting method (Liu et al., 2019) to ensure fair comparisons with previous work such as (Pang et al., 2022; Chen et al., 2024). Standard random scaling and random translation are applied for data augmentation during training and the input point clouds exclusively contain coordinate information, without additional normal information provided. The results are presented in Table 1. Our Point-PQAE yields better or comparable results compared to previous approaches.

**Few-shot Learning.** We conduct the few-shot learning classification on the ModelNet40 dataset to show the generalization ability of our method following the protocols in previous studies (Pang et al., 2022; Yu et al., 2022; Sharma & Kaul, 2020; Wang et al., 2021b). The few-shot learning experiments are conducted in the form of "$w$-way, $s$-shot" including four parts where $w \in \{5, 10\}$ represents the number of randomly selected classes and $s \in \{10, 20\}$ means the number of randomly selected samples for each $w$. Each part is conducted with 10 independent trials. The mean accuracy and standard deviation are reported in Table 2. Our Point-PQAE demonstrates superior performance compared to previous methods in few-shot learning, even surpassing cross-modal methods that utilize strong pre-trained teachers such as ACT and ReCon under FULL protocol. Specifically, Point-PQAE achieves 2.7%, 1.1%, 4.5%, 1.9% improvement under MLP-LINEAR protocol and shows enhanced performance over the self-construction method Point-MAE.

**Part Segmentation.** We conduct part segmentation experiments on the ShapeNetPart (Yi et al., 2016) to validate the effectiveness of our Point-PQAE. The ShapeNetPart contains 16,881 objects covering 16 categories. We sample 2048 points of each input point cloud for using following previous work (Yu et al., 2022; Pang et al., 2022) coming with dividing each of them into 128 point patches. We use the same segmentation head and utilize learned features from the $4th$, $8th$, and $12th$ layers of the Transformer block as in Point-MAE (Pang et al., 2022). We concatenate three levels of features. Then average pooling, max pooling, and upsampling are utilized to generate features for each point and an MLP is applied for label prediction. The results are reported in Table 3, which show that Point-PQAE achieves a comparable Inst.mIoU to the previous method which improves the from-scratch baseline by 1.4% and achieves 84.6% in Cls.mIoU.

Table 1: Classification accuracy (%) on ScanObjectNN and ModelNet40. The inference model parameters #P (M) are reported. Three variants are evaluated on ScanObjectNN and the accuracy obtained on ModelNet40 is reported for both 1K and 8K points. We compare methods using the ● plain Transformer architectures, *e.g.*Point-MAE (Pang et al., 2022), ○ hierarchical Transformer architectures and ○ dedicated architectures for 3D. The dagger($^\dagger$) denotes the baseline results reported from ReCon (Qi et al., 2023) which aligns augmentation with us and ReCon.

| Methods | #P | ScanObjectNN | | | ModelNet40 | |
|---|---|---|---|---|---|---|
| | | OBJ_BG | OBJ_ONLY | PB_T50_RS | 1K P | 8K P |
| *Supervised Learning Only* | | | | | | |
| ○PointNet (Qi et al., 2017a) | 3.5 | 73.3 | 79.2 | 68.0 | 89.2 | 90.8 |
| ○PointNet++ (Qi et al., 2017b) | 1.5 | 82.3 | 84.3 | 77.9 | 90.7 | 91.9 |
| ○DGCNN (Wang et al., 2019) | 1.8 | 82.8 | 86.2 | 78.1 | 92.9 | - |
| ○PointCNN (Li et al., 2018b) | 0.6 | 86.1 | 85.5 | 78.5 | 92.2 | - |
| ○SimpleView (Goyal et al., 2021) | - | - | - | 80.5±0.3 | 93.9 | - |
| ○MVTN (Hamdi et al., 2021) | 11.2 | 92.6 | 92.3 | 82.8 | 93.8 | - |
| ○PCT (Guo et al., 2021) | 2.88 | - | - | - | 93.2 | - |
| ○PointMLP (Ma et al., 2022) | 12.6 | - | - | 85.4±0.3 | 94.5 | - |
| ○PointNeXt (Qian et al., 2022a) | 1.4 | - | - | 87.7±0.4 | 94.0 | - |
| ○P2P-HorNet (Wang & Yoon, 2021) | - | - | - | 89.3 | 94.0 | - |
| *with Self-Supervised Representation Learning* (FULL) | | | | | | |
| ●Transformer (Vaswani et al., 2017) | 22.1 | 83.0 | 84.0 | 79.1 | 91.4 | 91.8 |
| ●Point-BERT (Yu et al., 2022) | 22.1 | 87.4 | 88.1 | 83.0 | 93.2 | 93.8 |
| ●MaskPoint (Liu et al., 2022) | - | 89.3 | 88.1 | 84.3 | 93.8 | - |
| ●Point-MAE (Pang et al., 2022) | 22.1 | 90.0 | 88.3 | 85.2 | 93.8 | 94.0 |
| ○Point-M2AE (Zhang et al., 2022) | 15.3 | 91.2 | 88.9 | 86.4 | **94.0** | - |
| ●Point-MAE$^\dagger$ (Pang et al., 2022) | 22.1 | 92.6 | 91.9 | 88.4 | 93.8 | 94.0 |
| ●**Point-PQAE** | 22.1 | **95.0** | **93.6** | **89.6** | **94.0** | **94.3** |
| Methods using cross-modal information and teacher models | | | | | | |
| ●Joint-MAE (Guo et al., 2023) | - | 90.9 | 88.9 | 86.1 | 94.0 | - |
| ●TAP (Wang et al., 2023) | 22.1 | 90.4 | 89.5 | 85.7 | 94.0 | - |
| ●ACT (Dong et al., 2022) | 22.1 | 93.3 | 91.9 | 88.2 | 93.7 | 94.0 |
| ○PointMLP+ULIP (Xue et al., 2023) | - | - | - | 89.4 | 94.5 | 94.7 |
| ○I2P-MAE (Zhang et al., 2023a) | 15.3 | 94.2 | 91.6 | 90.1 | 94.1 | |
| ●ReCon (Qi et al., 2023) | 43.6 | 95.2 | 93.6 | 90.6 | 94.5 | 94.7 |
| *with Self-Supervised Representation Learning* (MLP-LINEAR) | | | | | | |
| ●Point-MAE$^\dagger$ (Pang et al., 2022) | 22.1 | 82.8±0.3 | 83.2±0.2 | 74.1±0.2 | 90.2±0.1 | 90.7±0.1 |
| ●**Point-PQAE** | 22.1 | **89.3±0.3** | **90.2±0.4** | **80.8±0.1** | **92.0±0.2** | **92.2±0.1** |
| Methods using cross-modal information and teacher models | | | | | | |
| ●ACT (Dong et al., 2022) | 22.1 | 85.2±0.8 | 85.8±0.2 | 76.3±0.3 | 91.4±0.2 | 91.8±0.2 |
| ●ReCon (Qi et al., 2023) | 43.6 | 89.5±0.2 | 89.7±0.2 | 81.4±0.1 | 92.5±0.2 | 92.7±0.1 |
| *with Self-Supervised Representation Learning* (MLP-3) | | | | | | |
| ●Point-MAE$^\dagger$ (Pang et al., 2022) | 22.1 | 85.8±0.3 | 85.5±0.2 | 80.4±0.2 | 91.3±0.2 | 91.7±0.2 |
| ●**Point-PQAE** | 22.1 | **90.7±0.2** | **90.9±0.2** | **83.3±0.1** | **92.8±0.1** | **92.9±0.1** |
| Methods using cross-modal information and teacher models | | | | | | |
| ●ACT (Dong et al., 2022) | 22.1 | 87.1±0.2 | 87.9±0.4 | 81.5±0.2 | 92.7±0.2 | 93.0±0.1 |
| ●ReCon (Qi et al., 2023) | 43.6 | 90.6±0.2 | 90.7±0.3 | 83.8±0.4 | 93.0±0.1 | 93.4±0.1 |

**3D Scene Segmentation.** Semantic segmentation on large-scale 3D scenes is challenging, requiring models possessing a deep comprehension of contextual semantics and intricate local geometric relationships. We report semantic segmentation results on the S3DIS dataset (Armeni et al., 2016) in Table 3. Our Point-PQAE improves the from-scratch baseline by 2.0% and 1.4%, and outperforms the self-reconstruction method Point-MAE by 0.7% and 0.6% in mAcc and mIoU respectively.

## 4.3 ABLATION STUDY

Experiments are conducted to show the properties of our Point-PQAE. We report the classification results on three variants of ScanObjectNN. The experiment settings are aligned with Section 4.2.

**Relative Positional Embedding.** To analyze the effect of relative positional embedding in the positional query block, we conduct a series of experiments using different types of positional embeddings. We primarily consider two types of embeddings: Sin-Cos (as discussed in Section 3.2) and Learnable, along with a None group. Learnable embedding refers to utilizing the relative position information $\mathbf{RP}_{1\rightarrow2}$ and $\mathbf{RP}_{2\rightarrow1}$ as input and employing an MLP composed of two linear

Table 2: Few-shot classification results on ModelNet40. 10 independent trials are conducted in each experimental setting. The mean accuracy (%) and standard deviation are reported for each setting. The dagger($^\dagger$) denotes the baseline results reported from ReCon (Qi et al., 2023) which aligns augmentation with us and ReCon.

| Methods | 5-way | | 10-way | |
|---|---|---|---|---|
| | 10-shot | 20-shot | 10-shot | 20-shot |
| ∘DGCNN (Wang et al., 2019) | 31.6±2.8 | 40.8±4.6 | 19.9±2.1 | 16.9±1.5 |
| ∘OcCo (Wang et al., 2021a) | 90.6±2.8 | 92.5±1.9 | 82.9±1.3 | 86.5±2.2 |
| *with Self-Supervised Representation Learning* (FULL) | | | | |
| •Transformer (Vaswani et al., 2017) | 87.8±5.2 | 93.3±4.3 | 84.6±5.5 | 89.4±6.3 |
| •Point-BERT (Yu et al., 2022) | 94.6±3.1 | 96.3±2.7 | 91.0±5.4 | 92.7±5.1 |
| •MaskPoint (Liu et al., 2022) | 95.0±3.7 | 97.2±1.7 | 91.4±4.0 | 93.4±3.5 |
| •Point-MAE (Pang et al., 2022) | 96.3±2.5 | 97.8±1.8 | 92.6±4.1 | 95.0±3.0 |
| ∘Point-M2AE (Zhang et al., 2022) | 96.8±1.8 | 98.3±1.4 | 92.3±4.5 | 95.0±3.0 |
| •Point-MAE$^\dagger$ (Zhang et al., 2022) | 96.4±2.8 | 97.8±2.0 | 92.5±4.4 | 95.2±3.9 |
| •**Point-PQAE** | **96.9±3.2** | **98.9±1.0** | **94.1±4.2** | **96.3±2.7** |
| Methods using cross-modal information and teacher models | | | | |
| •Joint-MAE (Guo et al., 2023) | 96.7±2.2 | 97.9±1.9 | 92.6±3.7 | 95.1±2.6 |
| •TAP (Wang et al., 2023) | 97.3±1.8 | 97.8±1.9 | 93.1±2.6 | 95.8±1.0 |
| •ACT (Dong et al., 2022) | 96.8±2.3 | 98.0±1.4 | 93.3±4.0 | 95.6±2.8 |
| ∘I2P-MAE (Zhang et al., 2023a) | 97.0±1.8 | 98.3±1.3 | 92.6±5.0 | 95.5±3.0 |
| •ReCon (Qi et al., 2023) | 97.3±1.9 | 98.9±1.2 | 93.3±3.9 | 95.8±3.0 |
| *with Self-Supervised Representation Learning* (MLP-LINEAR) | | | | |
| •Point-MAE$^\dagger$ (Zhang et al., 2022) | 91.1±5.6 | 91.7±4.0 | 83.5±6.1 | 89.7±4.1 |
| •**Point-PQAE** | **93.0±4.6** | **96.8±1.9** | **89.0±5.2** | **93.5±3.8** |
| *with Self-Supervised Representation Learning* (MLP-3) | | | | |
| •Point-MAE$^\dagger$ (Zhang et al., 2022) | 95.0±2.8 | 96.7±2.4 | 90.6±4.7 | 93.8±5.0 |
| •**Point-PQAE** | **95.3±3.4** | **98.2±1.8** | **92.0±3.8** | **94.7±3.5** |

Table 3: Segmentation results including part segmentation on ShapeNetPart and semantic segmentation on S3DIS Area 5. We report mean intersection over union for all classes Cls.mIoU (%) and all instances Inst.mIoU (%) for part segmentation and mean accuracy mAcc (%) and IoU mIoU (%) of all categories for semantic segmentation.

| Methods | part seg. | | semantic seg. | |
|---|---|---|---|---|
| | Cls.mIoU | Inst.mIoU | mAcc | mIoU |
| ∘PointNet (Qi et al., 2017a) | 80.4 | 83.7 | 49.0 | 41.1 |
| ∘PointNet++ (Qi et al., 2017b) | 81.9 | 85.1 | 67.1 | 53.5 |
| ∘DGCNN (Wang et al., 2019) | 82.3 | 85.2 | - | - |
| ∘PointMLP (Ma et al., 2022) | 84.6 | 86.1 | - | - |
| *with Self-Supervised Representation Learning* | | | | |
| •Transformer (Vaswani et al., 2017) | 83.4 | 84.7 | 68.6 | 60.0 |
| ∘CrossPoint (Afham et al., 2022) | - | 85.5 | - | - |
| •Point-BERT (Yu et al., 2022) | 84.1 | 85.6 | - | - |
| •Point-MAE (Pang et al., 2022) | 84.2 | **86.1** | 69.9 | 60.8 |
| •**Point-PQAE** | **84.6** | **86.1** | **70.6** | **61.4** |
| Methods using cross-modal information and teacher models | | | | |
| •ACT (Dong et al., 2022) | 84.7 | 86.1 | 71.1 | 61.2 |
| •ReCon (Qi et al., 2023) | 84.8 | 86.4 | - | - |

layers with an activation function to map the 6-dimensional input to a D-dimensional output. The None group involves using randomly assigned embeddings to assess the effectiveness of RPE design. Table 4a displays the results of different positional embeddings. In the None group, the model cannot learn the relative positional information across two views, leading to a significant drop in accuracy. In addition to the type of RPE, we also experiment with Absolute Positional Embedding (APE) to compare its performance with our proposed RPE. The APE is unsuitable for our framework since it requires a shared coordinate system for both views, making it incompatible with the normalization and rotation used in our method. We incorporate APE into our Point-PQAE by removing these operations. The results in Table 4a show APE performs worse and support our claim.

**Data Augmentation.** After applying the random crop mechanism to the point cloud, we further perform data augmentation on the cropped views to generate new views. Data augmentation is

Table 4: Ablation study with Point-PQAE pre-training on ShapeNet. The classification results by accuracy (%) on three variants of ScanObjectNN are reported. Default settings are marked in gray.

<table>
<tr><td colspan="4">(a) Relative positional embedding</td><td colspan="4">(b) Data augmentation</td></tr>
<tr><td>Positional Embedding</td><td>OBJ_BG</td><td>OBJ_ONLY</td><td>PB_T50_RS</td><td>Data augmentation</td><td>OBJ_BG</td><td>OBJ_ONLY</td><td>PB_T50_RS</td></tr>
<tr><td>None</td><td>84.5</td><td>85.9</td><td>79.3</td><td>jitter</td><td>93.3</td><td>91.4</td><td>87.3</td></tr>
<tr><td>APE</td><td>92.3</td><td>91.0</td><td>87.7</td><td>scale</td><td>93.3</td><td>91.7</td><td>88.0</td></tr>
<tr><td>Learnable (RPE)</td><td>93.4</td><td>93.1</td><td>89.1</td><td>rotation</td><td>95.0</td><td>93.6</td><td>89.6</td></tr>
<tr><td>Sin-Cos Equation (7) (RPE)</td><td>95.0</td><td>93.6</td><td>89.6</td><td>scale&translate</td><td>92.8</td><td>91.7</td><td>88.1</td></tr>
<tr><td></td><td></td><td></td><td></td><td>rotation+scale&translate</td><td>93.8</td><td>92.9</td><td>89.1</td></tr>
</table>

<table>
<tr><td colspan="5">(c) 3D random crop mechanism</td><td colspan="5">(d) Effects of augmentations after random crop</td></tr>
<tr><td>Method</td><td>crop</td><td>OBJ_BG</td><td>OBJ_ONLY</td><td>PB_T50_RS</td><td>norm.</td><td>rotation</td><td>OBJ_BG</td><td>OBJ_ONLY</td><td>PB_T50_RS</td></tr>
<tr><td>Point-MAE</td><td>✗</td><td>92.6</td><td>91.9</td><td>88.4</td><td>✗</td><td>✗</td><td>92.8</td><td>92.1</td><td>88.2</td></tr>
<tr><td>Point-MAE</td><td>✓</td><td>92.9</td><td>92.1</td><td>88.8</td><td>✓</td><td>✗</td><td>93.3</td><td>92.6</td><td>87.9</td></tr>
<tr><td>Point-PQAE</td><td>✗</td><td>92.9</td><td>92.3</td><td>87.7</td><td>✗</td><td>✓</td><td>93.5</td><td>92.6</td><td>88.8</td></tr>
<tr><td>Point-PQAE</td><td>✓</td><td>95.0</td><td>93.6</td><td>89.6</td><td>✓</td><td>✓</td><td>95.0</td><td>93.6</td><td>89.6</td></tr>
</table>

crucial for generating diverse and decoupled views, so we conducted ablation studies on different augmentation methods. The results in Table 4b indicate that rotation performs best.

**Effectiveness of the Crop Mechanism.** Random crop has been shown to be highly effective in 2D self-supervised learning (SSL), particularly for contrastive learning (Grill et al., 2020; Chen et al., 2020; Bao et al., 2021), and is also vital for generative learning (He et al., 2022). However, its potential in 3D SSL remains largely unexplored. Even contrastive learning methods for point cloud understanding have not explored the power of it (Afham et al., 2022; Pang et al., 2023). To address this gap, we first introduce a custom-designed random crop mechanism for point cloud data, using it as the foundation for generating decoupled views. Without the crop augmentation, the two views will contain exactly the same shape though with normalization and rotation applied. **The model would only need to infer the augmentation rather than the inter-view relationship, which means the views are not decoupled.** Conversely, given two isolated cropped point clouds, followed by min-max normalization and rotation for further decoupling, there remain overlapping and non-overlapping parts, requiring the model to effectively encode the intra-view parts in order to infer the unknown inter-view points in our Point-PQAE. However, directly applying random crop as an additional augmentation to the self-reconstruction method (Point-MAE) does not improve the performance much, as shown in Table 4c. In other words, **it means our Point-PQAE fits crop mechanism better by harmoniously incorporating it into the decoupled views generation process.** The experiment results in Table 4c demonstrate the critical role of random crop in our method. We set minimum crop ratio $r_m = 1.0$ when removing the crop in our Point-PQAE.

**Analysis on Decoupled Views Generation.** While crop plays an important role in our method, relying solely on the random crop mechanism to generate new views would degenerate cross-reconstruction to simply reconstructing one part from another. This is similar to the block mask self-reconstruction used in Point-MAE (Pang et al., 2022), where visible blocks are used to reconstruct masked blocks, which is also trivial for pre-training. The only difference between them is that the cropped parts can overlap somewhat, whereas the block mask does not. What makes the cross-reconstruction different from block mask self-reconstruction are the subsequent augmentations including normalization and rotation that decouple the cropped parts. By normalizing cropped point clouds centered on the geometric centers, the coordinate systems of the two point clouds become isolated and independent from each other, with rotation further amplifying the variance between the two views, thus enabling the generation of two decoupled views. The importance of the augmentations applied after the random crop can be validated through experiments, as shown in Table 4d.

## 5 CONCLUSION

We have proposed Point-PQAE, a novel cross-reconstruction generative framework for self-supervised learning on 3D point clouds. In contrast to well-studied self-reconstruction schemes, Point-PQAE reconstructs one cropped point cloud via the other decoupled point cloud. To supply sufficient information to perform cross-reconstruction, we further propose a 3D relative positional embedding and corresponding position-aware query module. Compared to self-reconstruction, the cross-reconstruction task carefully designed by us is much more challenging for pre-training, promoting the learning of richer semantic representations during pre-training. This enables our proposed Point-PQAE to outperform previous single-modal self-reconstruction methods by a margin and to perform on par with or better than cross-modal methods.

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

APPENDIX

## A ADDTIONAL RELATED WORKS

Our Point-PQAE pioneers the cross-reconstruction paradigm in 3D point cloud self-supervised learning (SSL). There are two essential components in our proposed cross-reconstruction framework: 1) Two isolated/decoupled views, rather than two parts of the same instance that maintain a fixed relative relationship. 2) A model that achieves cross-view reconstruction using relative position information. To our knowledge, there are no similar previous methods in this domain. To better position our methodology, we compare it to similar SSL methods, including SiamMAE (Gupta et al., 2023) and CropMAE (Eymaël et al., 2025), proposed in the image domain. SiamMAE operates on pairs of randomly sampled video frames and asymmetrically masks them, utilizing the past frame to predict the masked future frame. CropMAE relies on image augmentations to generate two views, using one to reconstruct the other.

**Relation of SiamMAE and CropMAE to our Point-PQAE:** All are cross-reconstruction methods. Both SiamMAE and CropMAE have two essential components for cross-reconstruction-framework including multi-view (different frames sampled from one video for SiamMAE and isolated augmented images for CropMAE) and cross-view reconstruction model. They can be treated as cross-reconstruction methods, similar to our Point-PQAE.

**Difference of SiamMAE and CropMAE to our Point-PQAE: 1) Different domain:** SiamMAE and CropMAE focus on the 2D SSL domain. Our Point-PQAE is the first method for cross-reconstruction in the 3D SSL domain. **2) Asymmetric/symmetric reconstruction:** SiamMAE uses the past to predict the future, which is asymmetric. CropMAE performs asymmetric reconstruction and doesn't explore siamese cross-reconstruction. In contrast, our Point-PQAE is inherently symmetric, and the siamese loss brings a performance gain. **3) No relative information utilized:** SiamMAE and CropMAE do not incorporate relative information into training but rely on non-fully masking to guide the cross-reconstruction. The RPE adopted by our Point-PQAE provides explicit guidance, making training more stable and improving explainability. **4) No tuned-needed mask ratio exists in our framework.** There is a hyperparameter mask ratio that needs to be tuned in both SiamMAE and CropMAE, but this is not the case in our Point-PQAE framework.

## B MORE DISCUSSION ON THE RELATIVE POSITIONAL EMBEDDING AND POSITIONAL QUERY

**Related Relative Positional Embedding (RPE) Methods.** To better position the Relative Positional Embedding proposed by us for point cloud cross-view reconstruction, we discuss the difference between our RPE and other RPE methods. In the fields of Natural Language Processing (NLP) and 2D vision, RPE techniques have been widely adopted (Wu et al., 2021; Yang, 2019; Raffel et al., 2020). For instance, Rotary Positional Embedding (RoPE) (Su et al., 2024) is an emerging RPE technique gaining traction in the realm of large language models (LLMs). RoPE integrates rotational transformations to encode relative token positions, enabling more efficient extrapolation over unseen sequences. iRPE (Wu et al., 2021) first reviews existing relative position encoding methods, and then proposes new RPE methods dedicated to 2D images. The work (Qu et al., 2021) investigates the potential problems in Shaw-RPE and XL-RPE, which are the most representative and prevalent RPEs, and proposes two novel RPEs called LRHC-RPE and GCDF-RPE. Generally, in NLP and 2D vision, RPE captures the relative distances or orientations between tokens or pixels to enhance the model's capacity to understand relationships between paired elements. This approach often leads to improved generalization, especially when handling out-of-distribution data.

In contrast, our proposed RPE is designed with a view- or instance-based focus, rather than a token-based one. Rather than capturing relationships between individual tokens or pixels, our RPE encodes the positional relationships between two decoupled views. Our approach is not aimed at improving extrapolation or generalization. Instead, it is tailored to model the geometric and contextual information between different views to facilitate accurate cross-view reconstruction. This shift in focus

Table 5: Training details for pretraining and downstream fine-tuning.

| Config | ShapeNet | ScanObjectNN | ModelNet | ShapeNetPart | S3DIS |
|---|---|---|---|---|---|
| optimizer | AdamW | AdamW | AdamW | AdamW | AdamW |
| learning rate | 5e-4 | 2e-5 | 1e-5 | 2e-4 | 2e-4 |
| weight decay | 5e-2 | 5e-2 | 5e-2 | 5e-2 | 5e-2 |
| learning rate scheduler | cosine | cosine | cosine | cosine | cosine |
| training epochs | 300 | 300 | 300 | 300 | 60 |
| warmup epochs | 10 | 10 | 10 | 10 | 10 |
| batch size | 128 | 32 | 32 | 16 | 32 |
| drop path rate | 0.1 | 0.2 | 0.2 | 0.1 | 0.1 |
| number of points | 1024 | 2048 | 1024 | 2048 | 2048 |
| number of point patches | 64 | 128 | 64 | 128 | 128 |
| point patch size | 32 | 32 | 32 | 32 | 32 |
| augmentation | Rotation | Rotation | Scale&Trans | - | - |
| GPU device | RTX 3090 | RTX 3090 | RTX 3090 | RTX 3090 | RTX 3090 |

makes our RPE fundamentally distinct from existing RPE methods, highlighting the importance of carefully distinguishing our approach from other RPE techniques.

**Related Positional Query (PQ) Methods.** The positional query is also used in 2D self-supervised learning (SSL) and AI-generated content (AIGC). PQCL (Zhang et al., 2023b) pioneered the introduction of positional query, aiming to represent geometric relationships between multiple cropped views. PQDiff (Zhang et al., 2024) advanced this concept by devising a contiguous relative positional query module, applying it to image outpainting to achieve arbitrary location and contiguous expansion factor outpainting. Positional query has also found applications in 2D segmentation tasks. For example, DFPQ (He et al., 2023) generates positional queries dynamically by leveraging cross-attention scores from the previous decoder block and the positional encodings of the image features, which together enhance the effectiveness of semantic segmentation. Our method, however, distinguishes itself from these existing positional query approaches by focusing on the 3D world, which presents significantly greater complexity (one more dimension) and challenges compared to 2D image domains. By leveraging the obtained RPE to query the target view from the source view, our PQ technique successfully achieves decoupled view reconstruction.

## C  ADDITIONAL EXPERIMENTAL DETAILS

**Training Details.** We utilize ShapeNet (Chang et al., 2015) as our pre-training dataset, which comprises a curated collection of 3D CAD object models, featuring 51K unique models across 55 common categories. The pre-training process spans 300 epochs, employing a cosine learning rate schedule (Loshchilov & Hutter, 2016) starting at 5e-4, with a warm-up period of 10 epochs. We use the AdamW optimizer (Loshchilov & Hutter, 2017) and a batch size of 128. All experiments are conducted on a single GPU $i.e.$, RTX 3090 (24GB). For further training details including pre-training and finetuning, refer to Table 5. During the pre-training of our Point-PQAE on ShapeNet, we apply rotation to the input point cloud following ReCon (Qi et al., 2023), followed by generating decoupled views from the augmented point cloud.

**Finetuning Evaluation Protocol.** For classification tasks on ScanObjectNN and ModelNet40, as well as few-shot learning on ModelNet40, we adopt three evaluation protocols, following (Dong et al., 2022; Qi et al., 2023), to assess both the transferability of learned representations (FULL) and the quality of frozen features (MLP-LINEAR, MLP-3). The protocols are as follows:

(a) FULL: Fine-tuning the pre-trained model by updating both the backbone and the classification head.

(b) MLP-LINEAR: Fine-tuning by updating only the classification head, which consists of a single-layer linear MLP.

(c) MLP-3: Fine-tuning by updating only the parameters of a three-layer non-linear MLP classification head (which is structured the same as in FULL).

Table 6: Integrate PQ into distillation. Results on ScanobjectNN (%) are reported.

| Type | OBJ_BG | OBJ_ONLY | PB_T50_RS |
|---|---|---|---|
| Train from scratch | 83.0 | 84.0 | 79.1 |
| PQ distillation | 93.5 | 91.9 | 88.5 |

Table 7: Reconstruction loss function. The default setting is marked in gray.

| Loss Function | OBJ_BG | OBJ_ONLY | PB_T50_RS |
|---|---|---|---|
| cos | 90.5 | 89.8 | 85.2 |
| CD-$l1$ | 93.1 | 91.7 | 89.4 |
| CD-$l2$ | 95.0 | 93.6 | 89.6 |

Table 8: Siamese loss. The default setting is marked in gray.

| Loss Function | OBJ_BG | OBJ_ONLY | PB_T50_RS |
|---|---|---|---|
| $\mathcal{L}_{2\to1}$ | 93.4 | 92.4 | 89.2 |
| $\mathcal{L}_{2\to1} + \mathcal{L}_{1\to2}$ | 95.0 | 93.6 | 89.6 |

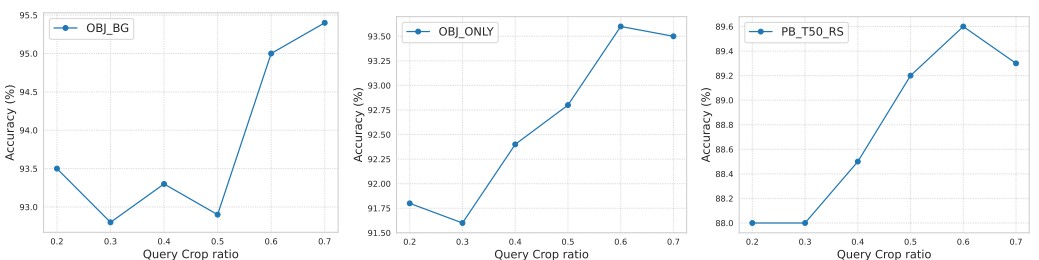

Figure 4: Ablation study on different minimum crop ratios $r_m$, where the results (%) of three variants: OBJ_BG, OBJ_ONLY, PT_T50_RS on ScanObjectNN are reported.

## D  ADDITIONAL ABLATION STUDY

**Integrate Positional Query (PQ) Scheme into Knowledge Distillation.** The knowledge distillation (Hinton et al., 2015) typically involves inputting the same instance into both the student model and the frozen teacher model, then maximizing the mutual agreement between their outputs to distill knowledge from the teacher to the student. Our positional query block can be seamlessly integrated into knowledge distillation, allowing for cross-view distillation rather than being confined to distillation within the same view. For example, view 1 is fed to the student, view 2 is fed to the teacher, and a positional query block is added after the backbone to model relative relations and recover the latent representation of view 2. We conduct experiments on distilling the pre-trained model ReCon (Qi et al., 2023), and the results are reported in Table 6, indicating that our PQ scheme successfully learns knowledge from the ReCon teacher and performs much better than the baseline. It shows that the PQ scheme can be easily utilized as a plug-in tool for knowledge distillation.

**Reconstruction Loss Function.** Table 7 shows the performance of Point-PQAE using different reconstruction loss functions: cosine similarity loss (cos), $l1$-form Chamfer distance (Fan et al., 2017) (CD-$l1$), and the $l2$-form Chamfer distance (CD-$l2$). The results show the CD-$l2$ is more suitable for Point-PQAE.

**Siamese Loss Function.** The generative pre-training task designed by us is naturally a siamese structure and we get the form of $\mathcal{L}_{cross} = \mathcal{L}_{2\to1} + \mathcal{L}_{1\to2}$ as stated in Section 3.3. We analyze the benefit of the siamese loss function by doing an ablation study with loss functions $\mathcal{L}_{cross} = \mathcal{L}_{2\to1} + \mathcal{L}_{1\to2}$ or $\mathcal{L}_{2\to1}$ only. The Table 8 presents the experiment results. It shows this siamese loss function contributes to the performance of our Point-PQAE and brings accuracy gain.

**Minimum Crop Ratio.** The minimum crop ratio $r_m$ is important for the proposed point cloud crop mechanism. We conduct experiments to analyze the effect of minimum random crop ratios on the performance. The results are reported in Figure 4. The results show that 0.6 is the best crop ratio for our Point-PQAE. When the ratio is too low, the model struggles to extract sufficient relevant

information from the cropped view for effective cross-reconstruction. Conversely, excessively high ratios make the task too straightforward, hindering the model from learning robust representations.

# E    LIMITATIONS AND FUTURE WORK

Point-PQAE is a novel cross-reconstruction generative learning paradigm that differs significantly from previous self-reconstruction methods, enabling more diverse and challenging pre-training. Point-MAE (Pang et al., 2022) pioneered the self-reconstruction paradigm in the point cloud self-supervised (SSL) learning field and variant optimizations are well explored, *e.g.*, cross-modal (Dong et al., 2022; Qi et al., 2023; Guo et al., 2023), masking strategy (Zhang et al., 2023a), and hierarchical architecture (Zhang et al., 2022; 2023a). **Compared to the well-studied self-reconstruction, cross-reconstruction remains significantly under-explored.** As the initial venture into cross-reconstruction, our Point-PQAE opens a new avenue for advancement in point cloud SSL. However, the model employs a vanilla transformer architecture and is constrained to single-modality knowledge. This architecture may not be optimally suited for cross-reconstruction tasks. Furthermore, the limited size of the available 3D point cloud datasets—due to the challenges in data collection—restricts the broader applicability of our single-modality approach. Future work could explore the integration of knowledge from additional modalities or the development of more efficient and appropriate architectures for the cross-reconstruction paradigm.

