# OpenReview forum: "Position-Query-Based Autoencoders for View Decoupled Cross Point Cloud Reconstruction and a Self-Supervised Learning Framework"
_ICLR.cc/2025/Conference — Submitted to ICLR 2025_

### Official Review · Reviewer_d4im · 2024-10-27

**Soundness:** 3
**Presentation:** 3
**Contribution:** 3
**Rating:** 8
**Confidence:** 3

**Summary:**

The work proposed Point-PQAE, a cross-reconstruction generative framework for self-supervised learning. The input point cloud is randomly cropped to generate two views. Then, the proposed positional query block extracts relative geometric relations to help the recontruction of one view from the other view. Extensive experiments are conducted and prove that Point-PQAE achieves good performance on multiple benchmarks.

**Strengths:**

1. Inspiring attempt. While other works focus on single-view reconstruction, this work attempts to recontruct one view from the other view, which is more diffucult. The experiment results show that this is a good attempt, and this work may inspire more researchers.
2. Thorough experiments. Multiple downstream tasks are performed, and the work acheives satisfying results.

**Weaknesses:**

N/A

**Questions:**

N/A

---

> ### Author Response · Authors · 2024-11-20
>
> Thank you very much for your time, concise comments, and valuable feedback. We are grateful for your recognition of the novelty and performance of our Point-PQAE framework.
>
> If you have any further questions or need additional clarifications, please feel free to reach out. We’d be happy to address any concerns.

---

> > ### Comment · Reviewer_d4im · 2024-11-26
> >
> > Thank you. After reading all the comments by other reviewers and your reply to them, it is clearer to me.
> > Good luck!

---

> > > ### Author Response · Authors · 2024-11-27
> > >
> > > Dear Reviewer d4im,
> > >
> > > We sincerely appreciate your feedback and the time you spent reading all the comments from other reviewers and our responses. We are delighted and inspired that you still highly recognize the value of our Point-PQAE.
> > >
> > > Most of the concerns have been adequately addressed (please refer to the global official comment for a brief summary). **We truly believe that the core concepts of our Point-PQAE and our responses are now clear, and we would be deeply grateful if you could reconsider your confidence score.** Your support would mean a great deal to us.
> > >
> > > Best regards, The Authors

---

### Official Review · Reviewer_nNPL · 2024-11-04

**Soundness:** 3
**Presentation:** 3
**Contribution:** 2
**Rating:** 5
**Confidence:** 5

**Summary:**

In this paper, the authors address the prevalent generative masking schemes used for pre-training 3D point cloud data by introducing a novel cross-reconstruction generative paradigm. This method significantly increases the pre-training challenge by reconstructing one cropped point cloud through another, contrasting with the simpler self-reconstruction methods. To facilitate this, a positional encoding is introduced to represent the 3D relative positions between the two point clouds.

**Strengths:**

1, The paper proposes a new framework to successfully bring cross-reconstruction to 3D generative field for point cloud self-supervised learning.

2. The writing is well-organized, clear, and easy to follow.

3. The experiments conducted are thorough and comprehensive.

**Weaknesses:**

1, The view construction approach does not make sense to me. In my opinion, using one part to reconstruct another within a block is merely a specific case of block usage.

2, The training framework lacks novelty. It introduces only a new relative position encoding (RPE) module and a cross-attention module in the decoder.

3, Relative position encoding is not a new concept [1], [2], and there is no specific adaptation for the 3D scenario in this work.

4, The results are questionable. Point-DAE [3] has shown that adding augmentation is highly effective for 3D MAE tasks, achieving 88.69 on PB-T50-RS. However, in the ablation study, using only absolute position embeddings achieves just 87.7. This raises doubts about the effectiveness of the proposed 'multi-view' learning and cross-attention. What if we simply add the learned relative embedding to Point-DAE instead?

[1] Rethinking and Improving Relative Position Encoding for Vision Transformer
[2] Explore Better Relative Position Embeddings from Encoding Perspective for Transformer Models
[3] Point-DAE: Denoising Autoencoders for Self-supervised Point Cloud Learning

**Questions:**

See weakness part.

---

> ### Author Response · Authors · 2024-11-20
> **Response to Q1-Q2**
>
> Thank you for taking the time to review our paper and for offering valuable questions and suggestions. We hope that our responses will address your concerns. Here is our clarification.
>
> > Q1. The view construction approach does not make sense to me. Using one part to reconstruct another within a block is merely a specific case of block usage.
>
> **It is important to distinguish between ‘parts’ and ‘views’ to understand the significance of decoupled view generation/construction.**
>
> In the related ablation study, which can be found in lines 519-529 of the main paper, we summarize the following: **Without independently applying augmentations after cropping, the relative relationships between the cropped ‘parts’ remain fixed**, and the cross-reconstruction becomes essentially a specific case of block masking in Point-MAE, where visible blocks are used to reconstruct the masked ones. **However, by performing view decoupling, the relative relationships between parts become more diverse, and we define these as ‘views’.** In this context, the relative relationships are dynamic, enabling a more challenging pre-training process and leading to improved performance on downstream tasks.
>
> This can be easily validated through experiment. When we replace the global random mask in Point-MAE with a block mask (where one part is reconstructed from another), the performance does not change significantly. **This is because block masking represents cross-part reconstruction, which does not benefit from multi-view learning:**
>
> Mask type|OBJ-BG|OBJ-ONLY|PB-T50-RS
> -|-|-|-
> Global random|92.6|91.9|88.4
> Block mask|93.1|92.4|87.8
>
> However, when switching from parts to views, the performance increases significantly, validating the effectiveness of decoupled view construction:
>
> Multi-view|Scale|Rotation|OBJ-BG|OBJ-ONLY|PB-T50-RS
> -|-|-|-|-|-
> ✗|✗|✗|92.3|91.0|87.7|
> ✓|✓|✗|**94.1**|92.1|88.2|
> ✓|✗|✓|93.6|92.4|**88.8**|
> ✓|✓|✓|93.8|**92.8**|**88.8**|
>
> Point-PQAE (APE) is used to eliminate confounding factors (e.g., RPE) and demonstrate the significance of distinguishing views from parts. For experimental details, please refer to the answer to Q4.2.
>
> In summary, we agree with your point that using one part to reconstruct another within a block is merely a specific case of block usage. **However, after decoupling the ‘parts,’ they are transformed into distinct ‘views’ that are no longer confined to the same block and exhibit greater diversity in our Point-PQAE.** This allows for more challenging pre-training, leading to improved performance on downstream tasks.
>
> > Q2. The training framework lacks novelty. It introduces only a new relative position encoding (RPE) module and a cross-attention module in the decoder.
>
> The main novelty and contribution of our proposed Point-PQAE is that **it is the first cross-reconstruction framework in the 3D SSL domain, enabling more challenging and informative pre-training** compared to previous self-reconstruction paradigms, resulting in significant performance improvements. To achieve this novel pre-training objective, we introduce the following components:
>
> - **Decoupled views generation**, which is novel in 3D SSL, as random cropping has never been used in previous pipelines.
> - **RPE generation**, which is also a novel approach and differs from existing RPE techniques. For details, please refer to Appendix B and the answer to Q3.
> - The cross-attention used in the **positional query block** is not new in itself, but **the use of RPE** as a query is novel in the context of 3D SSL.
>
> We believe the novelty of our work should not be judged solely by the number of new modules introduced, but by **considering the overall design and contributions of the entire pipeline.** For example, CropMAE [3] does not add new modules to the SiamMAE [4] model but still introduces innovation by transforming learning video into learning image transformations, thus alleviating the need for video datasets.

---

> ### Author Response · Authors · 2024-11-20
> **Response to Q3**
>
> > Q3. Relative position encoding is not a new concept [1], [2], and there is no specific adaptation for the 3D scenario in this work.
>
> Thank you for your question. **It's important to clarify that the form and objective of our proposed Relative Position Encoding (RPE) are fundamentally different from existing RPE methods, as discussed in Appendix B in our [paper](https://openreview.net/pdf?id=jZASmAlxp2).**
>
> Generally, in NLP and 2D vision, RPE captures the relative distances or orientations between tokens or pixels to enhance the model’s capacity to understand relationships between paired elements. There are some RPE methods proposed in image domain:
>
> - Wu et al. [1] first review existing relative position encoding methods, and then propose new RPE methods called iRPE dedicated to 2D images.
>
> - Qu et al. [2] investigate the potential problems in Shaw-RPE and
> XL-RPE, which are the most representative and prevalent RPEs, and propose two novel RPEs called LRHC-RPE and GCDF-RPE.
>
> These approaches generally aim to improve generalization and the model's understanding of relative information between different tokens, especially when handling out-of-distribution data. **We have added discussions of these two related works in Appendix B**; please refer to the [revised PDF](https://openreview.net/pdf?id=jZASmAlxp2).
>
> In contrast, **our proposed RPE is designed with a view- or instance-based focus, rather than a token-based one [1][2].** Rather than capturing relationships between paired tokens or pixels, our RPE encodes the positional relationships between two decoupled views, each consisting of tokens of point cloud patches.
>
> **Our approach is not aimed at improving extrapolation or generalization; rather, it is tailored to model the geometric and contextual information between different views to facilitate accurate cross-view reconstruction.** While they share the same name, the focus of our RPE is distinct from existing RPE techniques, and this shift in focus highlights the importance of distinguishing our approach from traditional RPE methods.
>
> **The specific adaptation of our RPE for the 3D scenario can be summarized as follows:** 1) view-based instead of token-based 2) focus on modeling cross-view information but not improving extrapolation or generalization. For a detailed explanation of the calculation of our proposed RPE, please refer to lines 247–260 in our [paper](https://openreview.net/pdf?id=jZASmAlxp2).

---

> ### Author Response · Authors · 2024-11-20
> **Response to Q4**
>
> > Q4.1 Point-DAE has shown that adding augmentation is highly effective for 3D MAE tasks, achieving 88.69 on PB-T50-RS.
>
> The performance improvement of Point-DAE over Point-MAE comes not only from the augmentation but also from **the additional pre-training objective** (Section III.C in the Point-DAE paper). The comparison between Point-MAE and Point-DAE is as follows:
>
> Method|pre-training objective|pre-training aug|finetuning aug|OBJ-BG|OBJ-ONLY|PB-T50-RS
> -|-|-|-|-|-|-
> Point-MAE $^\dagger$|Mask autoencoding|Rotation|Rotation|92.6|91.9|88.4|
> Point-DAE|Center predicting+Mask autoencoding|Affine|Rotation|93.6|92.8|88.7|
>
> $^\dagger$ results reported from ReCon [5].
>
> The improvement of Point-DAE over Point-MAE is somewhat marginal, and the additional center-predicting objective has been shown to improve performance (TABLE X in the Point-DAE paper). Therefore, **it remains uncertain to what degree the affine transformation benefits 3D MAE tasks.**
>
> > Q4.2 The results are questionable. In the ablation study, using only absolute position embeddings achieves just 87.7 while Point-DAE achieves 88.69. This raises doubts about the effectiveness of the proposed 'multi-view' learning and cross-attention.
>
> Thank you for your careful review. Sorry for misleading you due to insufficient experiment statement in the paper. During the decoupled views generation, **min-max normalization (which decentralizes and then scales to [-1, 1]) and rotation** are applied to the cropped point clouds. **The decentralization isolates the coordinate systems of the two views, making APE no longer applicable.** Therefore, when experimenting with Point-PQAE (APE), we removed these augmentations after the random crop, which turns Point-PQAE (APE) into **a multi-part/multi-block learning task, rather than multi-view learning.** For the difference between `part` and `view`, please check the answer to Q1.
>
> Isolatedly augmenting different parts is necessary for transforming parts into views. **For clarity, since decentralization makes APE not applicable, we perform an ablation by adding scale and rotation (or not) to the cropped point clouds using Point-PQAE (APE) to explore the effectiveness of multi-view learning.** The results are shown as follows:
>
> Multi-view|Scale|Rotation|OBJ-BG|OBJ-ONLY|PB-T50-RS
> -|-|-|-|-|-
> ✗|✗|✗|92.3|91.0|87.7|
> ✓|✓|✗|**94.1**|92.1|88.2|
> ✓|✗|✓|93.6|92.4|**88.8**|
> ✓|✓|✓|93.8|**92.8**|**88.8**|
> Point-DAE|-|-|93.6|92.8|88.8|
>
> Though Point-DAE utilizes more powerful augmentation and even adopts an additional pre-training target, the Scale+Rotation Point-PQAE (APE) outperforms it. **This demonstrates the effectiveness of the proposed multi-view learning.**
>
> > 4.3 What if we simply add the learned relative embedding to Point-DAE instead?
>
> The relative positional embedding (RPE) consists of two parts: the geometry center difference between two views and the centers of point patches within each view. **RPE is used to model the relationship between two decoupled views that lie in isolated coordinate systems, enabling cross-view reconstruction.**
>
> **The RPE is not necessary and not feasible for all self-reconstruction methods, including Point-MAE [6] and Point-DAE.** For example, in Point-DAE with block masking, the visible and masked blocks remain within the same coordinate system and retain a fixed relative relationship, even without isolated augmentation. In such cases, directly using APE is sufficient and more appropriate than using RPE.
>
> As demonstrated by our proposed Point-PQAE, we can transform self-reconstruction methods like Point-MAE (block-mask) into cross-reconstruction by introducing view decoupling, RPE generation, and positional query.
>
> Reference
>
> [1] Wu K, Peng H, Chen M, et al. Rethinking and improving relative position encoding for vision transformer[C]//ICCV 2021.
>
> [2] Qu A, Niu J, Mo S. Explore better relative position embeddings from encoding perspective for transformer models[C]//EMNLP 2021.
>
> [3] Eymaël A, Vandeghen R, Cioppa A, et al. Efficient Image Pre-Training with Siamese Cropped Masked Autoencoders[C]//ECCV 2024.
>
> [4] Gupta A, Wu J, Deng J, et al. Siamese masked autoencoders[J]. NeurIPS 2023.
>
> [5] Qi Z, Dong R, Fan G, et al. Contrast with reconstruct: Contrastive 3d representation learning guided by generative pretraining[C]//ICML 2023.
>
> [6] Pang Y, Wang W, Tay F E H, et al. Masked autoencoders for point cloud self-supervised learning[C]//ECCV 2022.

---

> ### Comment · Reviewer_nNPL · 2024-11-22
>
> Thanks for the detailed response. However, I still have questions about the results and the technical novelty.
>
> 1, Regarding Table 1 in your reply, I don't believe the improvement gained by adding the multi-view component can be considered significant. For the PB-T50-RS, it shows only a 0.4% improvement. Adding cross-attention and RPE contributes just a further 0.8% increase. This feels incremental at best. If the authors were to start from this baseline and conduct an ablation study for each component, I think the improvement introduced by each would be similarly incremental. I hope the author can add the claim that the baseline method is Point-MAE reported in ReCon.
>
> Could you please conduct an ablation study starting from this baseline to evaluate the effectiveness of each component?
>
> 2, Additionally, I disagree with the SOTA claim in this paper. Methods like Point-FEMAE [1] demonstrate higher performance compared to this work. Also, the **performance compared to the baseline cannot be said to be significant (1.2 in PB T50 RS), even lower than ReCon results with simple 2D contrastive learning and without foundation models**.
>
> 3, Lastly, I did not observe any notable modifications in the RPE that would distinguish it from other implementations. From my perspective, this paper lacks technical novelty. It introduces a new augmentation method, implements existing RPE techniques, and employs commonly used cross-attention mechanisms, ultimately achieving only incremental improvements over the baseline (as reported in Point-MAE results in ReCon).
>
>
> [1] Towards compact 3d representations via point feature enhancement masked autoencoders.

---

> > ### Author Response · Authors · 2024-11-22
> > **Quick response to Q1**
> >
> > Thank you for your detailed response. It seems there remain significant misunderstandings regarding our proposed Point-PQAE on your side. We'd like to clarify this for you.
> >
> > > Q1.1 Regarding Table 1 in your reply, I don't believe the improvement gained by adding the multi-view component can be considered significant. For the PB-T50-RS, it shows only a 0.4% improvement. Adding cross-attention and RPE contributes just a further 0.8% increase. This feels incremental at best.
> >
> > There are three newly proposed components: 1) Decoupled views generation, 2) RPE generation, 3) Positional query (cross-attention).
> >
> > **It seems that you misunderstand the existing components for each setting, for example, the positional query which utilizes cross-attention actually is applied to all settings.** It's important for us to further clarify on the results in the table 1:
> >
> >
> >
> > |Method|Multi-view|Translate|Scale|Rotation|OBJ-BG|OBJ-ONLY|PB-T50-RS|component|
> > -|-|-|-|-|-|-|-|-
> > |Point-PQAE (APE)|✗|✗|✗|✗|92.3|91.0|87.7|3)
> > |Point-PQAE (APE)|✓|✗|✓|✗|94.1|92.1|88.2|1) 3)
> > |Point-PQAE (APE)|✓|✗|✗|✓|93.6|92.4|88.8|1) 3)
> > |Point-PQAE (APE)|✓|✗|✓|✓|93.8|92.8|88.8|1) 3)
> > |Point-PQAE (RPE)|✓|✓|✓|✓|95.0|93.6|89.6|1) 2) 3)
> >
> > Note that combining Translate (for decentralization) and Scale (to [-1, 1]) means min-max normalization (see the answer to the original Q4.2).
> >
> > Actually, **the results in the table exactly show the effectiveness of our proposed cross-reconstruction framework** over self-reconstruction or the multi-view paradigm over the single-view paradigm. It can be clearly observed that, with fixed relative relations, the cross-parts Point-PQAE (APE) achieves rather low performance (Line 1 in the table). By simply adding the following decoupling augmentations, the views possess dynamic relative relations between them and achieve significant performance improvement.
> >
> > Regarding the reason why decentralization (Translate) and RPE are added at the same time, you may check the answer to Q3.1. **We kindly remind you that our RPE models two isolated coordinate systems rather than modeling paired token relations (which is the aim of all other existing RPEs).**
> >
> > For the statement, `Adding cross-attention and RPE contributes just a further 0.8% increase. This feels incremental at best,` it should be corrected to `adding RPE and decentralization improves performance by 1.2, 0.8, and 0.8%` because the cross-attention exists in every experiment. For the statement, `multi-view shows only a 0.4% improvement over single-view,` it should also be corrected to `multi-view shows 1.5, 1.8, and 1.1% improvement over single-view by simply adding decoupled views generation (Line 4 vs Line 1) without RPE added for fair comparison.`
> >
> > > Q1.2 If the authors were to start from this baseline and conduct an ablation study for each component, I think the improvement introduced by each would be similarly incremental. I hope the author can add the claim that the baseline method is Point-MAE reported in ReCon. Could you please conduct an ablation study starting from this baseline to evaluate the effectiveness of each component?
> > Actually, apart from the performance of Point-MAE, which is reported from ReCon for convenience and aligns all hyperparameters and augmentations with ours, all other ablations are conducted on the same model architecture as Point-MAE (apart from one additional cross-attention layer for the positional query).
> >
> > We do not fully understand what `this baseline` means. If `this baseline` refers to the original architecture of Point-MAE:
> >
> > We remind you that the Point-MAE (as reported by ReCon) performs much better than the original Point-MAE on the ScanObjectNN due to the augmentations, i.e., Scale & Translate (Pre-train) + Scale & Translate (Finetune) for the original Point-MAE vs. Rotation (Pre-train) + Rotation (Finetune) for Point-MAE in ReCon.
> >
> > We built our code upon the code of ReCon, removing all other components apart from the modules in the original Point-MAE to ensure easier alignment of hyperparameters and augmentations with Point-MAE (baseline). **Thus, please rest assured that we only report the Point-MAE performance from ReCon to ensure fair comparison, and we do not adopt any components from ReCon that are not present in Point-MAE into our pipeline.** All ablations are conducted fairly.
> >
> > If you would like to reconsider the rating and check our code, an anonymous link can also be provided without issue.
> >
> > Regarding the incremental performance increase you mentioned, we have addressed this in our response to Q1.1 by further clarifying some points that you may have misunderstood.

---

> > ### Author Response · Authors · 2024-11-22
> > **Quick response to Q3**
> >
> > > Q3.1 Lastly, I did not observe any notable modifications in the RPE that would distinguish it from other implementations.
> >
> > **There remain significant misunderstandings on your side regarding the proposed RPE.** Here, we would like to re-emphasize that our proposed RPE consists of two components: **Relative Coordinates** (Geometry View Center 1 minus Geometry View Center 2) and regular Positional Encoding.
> >
> > 2D regular PE (Positional Encoding) typically requires the **index** of a patch or token as input to generate positional embeddings. For Point Cloud Data, previous PE approaches use the center's xyz coordinates as input. In contrast, our method additionally requires the **relative coordinates** between two views because view 1 **does not have information about** the position of view 2.
> >
> > We think "Relative Positional Encoding" is an appropriate name for our proposed method, which combines relative coordinates, patch centers, and then calculate with regular Positional Encoding (sinusoid positional encoding), **but we unintentionally chose the same name as some well-established existing RPEs.**
> >
> > For the existing RPEs that coincidentally share the same name, they model the relations between paired tokens, such as RoPE, which is fundamentally different from regular PE. For example, in a sentence with three words, the regular PE would be represented as 0, 1, 2, while the RPE would be modeled as 1-0, 2-0, 0-1, 2-1, 1-2, and 0-2. **In contrast, our proposed RPE combines the differences between two view centers and patch centers, and then calculates positional embeddings in the same way as regular PE, without considering paired token relations as existing RPEs.**
> >
> >
> > > Q3.2 It introduces a new augmentation method, implements existing RPE techniques, and employs commonly used cross-attention mechanisms, ultimately achieving only incremental improvements over the baseline (as reported in Point-MAE results in ReCon).
> >
> > **For the performance gain significance, we re-emphasize that our Point-PQAE outperforms Point-MAE on most benchmarks (see the answer to Q2.3),** e.g., our Point-PQAE outperforms Point-MAE by 1.8%, 6.7%, and 4.4% on FULL, Mlp-Linear, and Mlp-3 by averaging the performance gain across three variants of ScanObjectNN.
> >
> > For the novelty, the three main proposed components are novel themselves (see the answer to the original Q3), and we have stated that **we believe the novelty of our work should not be judged solely by the number of new modules introduced but by considering the overall design and contributions of the entire pipeline.**
> >
> > The main novelty and contribution of our proposed Point-PQAE is that **it is the first cross-reconstruction framework in the 3D SSL domain, enabling more challenging and informative pre-training compared to previous self-reconstruction paradigms, resulting in significant performance improvements.**
> >
> > If judging primarily by the number of modules added, the **NeurIPS 2023 (oral) SiamMAE** adopts a similar architecture to the MAE, **leveraging different frames in the videos and implementing cross-attention on the decoder side.** The only differences from the MAE lie in SiamMAE's domain focus, the use of two frames, and the application of commonly used cross-attention mechanisms, which might be interpreted as incremental improvements depending on the perspective.
> >
> > Similarly, for CropMAE (ECCV 2024), it applies different frames as a form of image augmentation. **It might appear that this approach introduces relatively limited new components.**
> >
> > Therefore, we kindly suggest evaluating the novelty of a paper more holistically, considering not only the added modules but also the motivation, overall pipeline and its contributions.

---

> ### Author Response · Authors · 2024-11-22
> **Quick response to Q2**
>
> > Q2.1 Additionally, I disagree with the SOTA claim in this paper. Methods like Point-FEMAE [1] demonstrate higher performance compared to this work.
>
> Thank you for mentioning the related work that is currently uncited.
>
> We will first introduce Point-FEMAE briefly and then compare it with our Point-PQAE to distinguish the two methods:
> - **Motivation:**
>   - Motivated by Point-MAE easily suffering from limited 3D representations with a global random mask input (Point-FEMAE)
>   **vs.**
>   - Motivated by the strengths of the two-view learning in other domains and pioneered a brand new cross-reconstruction paradigm in the point cloud SSL domain (ours).
>
> - **Method:**
>   - It adds an additional Local Enhancement Module (LEM) to each block in the Transformer, which is retained during finetuning **(27.4M vs. 22.1M (ours); finetune time: 30s/epoch vs. 10s/epoch (ours) for ScanObject (OBJ_BG))**.
>   - It utilizes two self-reconstruction branches, which make pre-training more complex **(27.2h vs. 19.6h (ours))**.
>
> **The data shows that we still outperform it in some metrics.** We display the performance comparison:
> ScanObjectNN|OBJ-BG|OBJ-ONLY|PB-T50-RS
> -|-|-|-
> Point-FEMAE (self-reconstruction)|**95.2**|93.3|**90.2**
> Point-PQAE (ours cross-reconstruction)|95.0|**93.6**|89.6
>
> ModelNet FewShot | 5w10s | 5w20s | 10w10s | 10w20s
> -|-|-|-|-
> Point-FEMAE |**97.2±1.9**|98.6±1.3|94.0±3.3|95.8±2.8
> Point-PQAE |96.9±3.2| **98.9±1.0**| **94.1±4.2**| **96.3±2.7**
>
> We will add the discussion and comparison with Point-FEMAE to the main paper. **Additionally, regarding the additional parameters and computational requirements for Point-FEMAE during finetuning, the comparison may be somewhat unfair.** As we discussed in Appendix E:
>
> > As the initial venture into cross-reconstruction, our Point-PQAE opens a new avenue for advancement in point cloud SSL. However, the model employs a vanilla transformer architecture and is constrained to single-modality knowledge. This architecture may not be optimally suited for cross-reconstruction tasks.
>
> > Q2.2 Also, the performance compared to the baseline cannot be said to be significant (1.2 in PB T50 RS).
>
> It's inappropriate to judge the performance based on merely one benchmark. **We demonstrate this issue with data by displaying the baseline (Point-MAE), our method, and the leading cross-modal method (ReCon):**
>
> ScanObjectNN (FULL)|OBJ-BG|OBJ-ONLY|PB-T50-RS
> -|-|-|-
> Point-MAE (single-modal)|92.6|91.9|88.4|
> Point-PQAE (ours single-modal)|95.0|**93.6**|89.6
> ReCon (cross-modal SOTA)|**95.2**|**93.6**|**90.6**
>
> ScanObjectNN (Mlp-Linear)|OBJ-BG|OBJ-ONLY|PB-T50-RS
> -|-|-|-
> Point-MAE |82.8±0.3|83.2±0.2|74.1±0.2
> Point-PQAE (ours)|89.3±0.3|**90.2±0.4**|80.8±0.1
> ReCon |**89.5±0.2**|89.7±0.2|**81.4±0.1**
>
> ScanObjectNN (Mlp-3)|OBJ-BG|OBJ-ONLY|PB-T50-RS
> -|-|-|-
> Point-MAE |85.8±0.3|85.5±0.2|80.4±0.2
> Point-PQAE (ours)|**90.7±0.2**|**90.9±0.2**|83.3±0.1
> ReCon |90.6±0.2|90.7±0.3|**83.8±0.4**
>
> ModelNet | 1k | 8k | 5w10s | 5w20s | 10w10s | 10w20s
> -|-|-|-|-|-|-
> Point-MAE |93.8|94.0|96.4±2.8|97.8±2.0|92.5±4.4|95.2±3.9
> Point-PQAE (ours)|94.0|94.3|96.9±3.2| **98.9±1.0**| **94.1±4.2**| **96.3±2.7**
> ReCon |**94.5**|**94.7**|**97.3±1.9**|**98.9±1.2**|93.3±3.9|95.8±3.0
>
> ShapeNetPart & S3DIS|Cls.mIoU (part)|Inst.mIoU| Cls.mIoU (semantic)|Inst.mIoU|
> -|-|-|-|-
> Point-MAE (single-modal)|84.2|86.1|69.9|60.8
> Point-PQAE (ours single-modal)|84.6|86.1|**70.6**|**61.4**
> ReCon (cross-modal SOTA)|**84.8**|**86.4**|-|-
>
> In most of the benchmarks, **our Point-PQAE outperforms the baseline significantly** just as the more complex cross-modal SOTA method ReCon does.
>
> For the ScanObjectNN (FULL), our Point-PQAE outperforms Point-MAE by 1.2 on the PB-T50-RS variant, but it's hard to define whether 1.2 is `significant` and judge significance from just one variant. Through multiple benchmarks, **whether our Point-PQAE outperforms Point-MAE by 1.8%, 6.7%, and 4.4% on FULL, Mlp-Linear, and Mlp-3, respectively, by averaging the performance gain across three variants of ScanObjectNN is significant? We think the answer is obviously yes.**
>
> > Q2.3 The PB-T50-RS (89.6) of Point-PQAE performs even lower than ReCon results with simple 2D contrastive learning and without foundation models.
>
> The results related to 2D contrastive learning in Table 4 in the paper of ReCon can be summarized as:
>
> | Reconstruction |Contrastive (Image) | PB-T50-RS|
> |-|-|-|
> ✗|✓|86.02
> √|✓|90.18
> Point-PQAE (ours)||89.6
>
> If you mean that simple 2D contrastive learning without MAE can outperform our Point-PQAE, there may be some misunderstanding because it's 89.6 (ours) vs. 86.0.
>
> If you mean that Point-PQAE (89.6) underperforms MAE + Image contrastive, **please note that ReCon (MAE + Image) adopts a more complex pipeline** than ours, including a pre-trained image teacher (ViT-B), paired point-cloud data, longer pre-training time (37.7h for MAE + Image + Text vs. 19.6h for ours), and more trainable parameters during pre-training (140.9M for MAE + Image + Text vs. 29.6M for ours).

---

> > ### Comment · Reviewer_nNPL · 2024-11-23
> > **Reviewer Reply to Q2**
> >
> > For the parameters comparison (27.4M vs. 22.1M), I believe the difference between the proposed method and Point-FEMAE can still be considered a fair comparison, as both approaches modify the encoder. Furthermore, in downstream tasks, the most important and challenging results are typically seen in the PB-T50-RS split. However, in this particular split, the proposed method demonstrates only limited improvement. In ReCon, even without foundation models, the performance still reaches 89.7.
> >
> > Additionally, the key motivation of this paper is to highlight the effectiveness of multi-view reconstruction. However, when considering Point-PQAE (APE), which combines components 1) and 3), the addition of multi-view reconstruction only results in limited performance gains, achieving 94.1, 92.1, and 88.2. This raises significant doubts about the effectiveness of the proposed multi-view component.
> >
> > If the author directly adds component 3) to Point-MAE in ReCon, what are the resulting metrics? How do they compare when further incorporating RPE? As indicated by the experiments presented above, the main improvement appears to be primarily introduced by the RPE. I believe that adding any state-of-the-art learnable positional embedding would lead to performance improvements, regardless of whether RPE is used. Therefore, I strongly question the motivation behind attempting to demonstrate the effectiveness of multi-view reconstruction, as the experimental results are not compelling enough to substantiate this claim.

---

> ### Comment · Reviewer_nNPL · 2024-11-23
> **Reviewer Reply to Q1**
>
> From my perspective, the author uses Point-MAE (from ReCon, leveraging rotation augmentation) as the baseline. As mentioned earlier, it would be helpful to provide ablation studies based on Point-MAE with rotation augmentation, as this technique is now commonly adopted in most 3D self-supervised pre-training methods. It would also be preferable if the baseline starts with performance metrics of 92.6, 91.9, and 88.4.
>
> In the first row of the results, adding component 3) improves the baseline from the original MAE (90.02, 88.29, 85.18) to 92.3, 91.0, and 87.7. However, after introducing rotation augmentation, as shown in the third row, the improvement compared to MAE with rotation augmentation is marginal—only from 92.6, 91.9, and 88.4 to 93.6, 92.4, and 88.8. This suggests that the proposed component 3) provides only incremental gains.
>
> Furthermore, in the original paper (Table 4b), using only rotation augmentation achieves the best performance. In contrast, in this study, combining Translate, Scale, and Rotation is required to achieve optimal results. This discrepancy raises questions about the reliability of the results and whether the proposed approach truly provides a significant improvement.

---

> > ### Author Response · Authors · 2024-11-23
> > **Response to `Reviewer Reply to Q1` part 2**
> >
> > > Q1.3 Furthermore, in the original paper (Table 4b), using only rotation augmentation achieves the best performance. In contrast, in this study, combining Translate, Scale, and Rotation is required to achieve optimal results. This discrepancy raises questions about the reliability of the results and whether the proposed approach truly provides a significant improvement.
> >
> > This question has been addressed in Q1.2. We decompose min-max normalization and rotation, experimenting with different combinations to demonstrate that **the framework only works effectively when the components are sufficiently decoupled.**
> >
> > The use of Translate and Scale might appear misleading because they are not random augmentations. **Instead, they are components decomposed from the min-max normalization process**: decentralization (Translate) and scaling to [-1, 1] (Scale). **This operation is analogous to recentering and resizing an image in the 2D domain to ensure uniformity after cropping.**
> >
> > In other words, random rotation is the only additional augmentation for further decoupling, and it can be replaced by other compositions. However, **min-max normalization is a must just like in 2D domain and cannot be replaced.** We just conducted ablation studies to show that **the framework only works effectively with sufficient decoupling of these components.** by eliminating the min-max norm (in line 3 experiment result.)
> >
> > **Min-max normalization + Rotation is precisely our method (result line 5, not line 3).** Therefore, there is no discrepancy between the results in the rebuttal and those reported in Table 4b of the original paper.

---

> ### Comment · Reviewer_nNPL · 2024-11-23
> **Reviewer Reply to Q3**
>
> Regarding the relative positional embedding (RPE) component, the input should indeed be tailored to the specific context: at the token level in NLP, at the pixel position level in images, and at the point cloud coordinate level in 3D. My point about "directly using" RPE is whether any specific RPE structure was uniquely designed for this paper. Moreover, to my knowledge, RPE is not specifically designed for the multi-view scenario; it should also be effective for standard MAE cases without multi-view involvement. Given that the main improvement in this paper seems to come from the use of RPE, it weakens the overall motivation for emphasizing multi-view reconstruction.
>
> Regarding performance gains, as I previously mentioned, the most critical benchmark is the PB-T50-RS split, where the observed improvement remains limited.

---

> ### Author Response · Authors · 2024-11-23
>
> Thank you for your quick response and responsible attitude towards the review. **However, we believe you still have significantly misunderstood both the paper and the rebuttal.** The concerns you raised were addressed repeatedly in both, **but the lengthy response may have obscured the main points.** To clarify, we will restate key points concisely and strongly encourage you to re-evaluate the techniques and experiments in **Point-PQAE**, as there is a misunderstanding of the experimental setup on your side.

---

> ### Author Response · Authors · 2024-11-23
> **Response to `Reviewer Reply to Q1` part 1.**
>
> > Q1.1 From my perspective, the author uses Point-MAE (from ReCon, leveraging rotation augmentation) as the baseline. As mentioned earlier, it would be helpful to provide ablation studies based on Point-MAE with rotation augmentation, as this technique is now commonly adopted in most 3D self-supervised pre-training methods. It would also be preferable if the baseline starts with performance metrics of 92.6, 91.9, and 88.4.
>
> **We start exactly with Point-MAE using rotation augmentation as baseline** and do not adopt the original Point-MAE augmentations (Scale+Translate). We do not compare our Point-PQAE with Point-MAE (Scale+Translate) in any ablations.
>
> For convenience, we use results provided by ReCon, a classic paper in the 3D SSL domain, as it includes all the required results. Reporting these results for Point-MAE (rotation) simplifies our analysis, **as they exactly align with our augmentations and hyperparameters**.
>
> > Q1.2 In the first row of the results, adding component 3) improves the baseline from the original MAE (90.02, 88.29, 85.18) to 92.3, 91.0, and 87.7. However, after introducing rotation augmentation, as shown in the third row, the improvement compared to MAE with rotation augmentation is marginal—only from 92.6, 91.9, and 88.4 to 93.6, 92.4, and 88.8. This suggests that the proposed component 3) provides only incremental gains.
>
> We apologize for any confusion regarding Translate, Scale, and Rotation. **After cropping, we apply min-max normalization, which includes decentralization (Translate) and rescaling to [-1, 1] (Scale), as an initial step to decouple the two views.** The subsequent augmentation, designed to introduce further variance between views, is **Random Rotation**.
>
> This is **not random** Translate and Scale but rather min-max normalization (decentralization and scaling to [-1, 1])—**an operation analogous to the image domain**, where images are cropped, recentered, and rescaled to a fixed resolution (e.g., 224x224). For further clarification, we recommend you reviewing lines 178-204 of the paper.
>
> To clarify:
>
> **Random Rotation is the only additional augmentation for further decoupling,** which can be replaced with other compositions, but **min-max normalization is essential and cannot be replaced.**
>
> **Regarding the results:**
> - It's expected that **Point-PQAE (APE) Line 1** underperforms Point-MAE (Rotation):
>     - We emphasize that we **do not use the original MAE (90.02, 88.29, 85.18) as a baseline** in any experiments. Line 1 does not improve upon the original Point-MAE, as we do not start from it.
>     - Line 1 setting: crop two point clouds independently with $r_m=0.6$ without following view decoupling operations (min-max norm and rotation)
>     - As a result, the two point clouds overlap significantly and maintain fixed relative relations.
>     - This makes reconstructing one part from the other relatively trivial in the `single-view` setting—**even more trivial than in Point-MAE, which reconstructs non-overlapping parts.**
>     - Hence, Point-PQAE (APE) Line 1 underperforms Point-MAE (Rotation).
> - **It’s incorrect to directly compare each ablation result to Point-MAE.**
>     - Point-PQAE uses overlapping parts, whereas Point-MAE uses non-overlapping masked and visible parts.
>     - Similarly, in the 2D domain, CropMAE uses overlapping views while MAE uses non-overlapping masked and visible parts.
>     - **The appropriate comparison baseline is Point-PQAE (APE) Line 1, not Point-MAE.**
> - **It’s also incorrect to view Row 3 as the sole comparison between Point-PQAE and Point-MAE (Rotation).**
>     - Min-max normalization is an essential step for initial view decoupling, analogous to rescaling to 224x224 after cropping in the 2D domain.
>     - Please focus on the comparison of min-max normalization + rotation (Line 5) vs. Point-MAE (Rotation).
>
> - We decompose min-max normalization and rotation into three operations for a complete ablation study, as decentralization alone renders APE infeasible.
> - We try different compositions to show that **only with sufficiently decoupled parts does the framework work.** See Line 5 versus other lines.
>
> ### Lastly, we reiterate that Translate and Scale in the result refer to min-max normalization, not random augmentations. This is a critical operation post-cropping, analogous to rescaling after cropping in the 2D domain.
>
> **We kindly remind you of this again.**

---

> ### Author Response · Authors · 2024-11-23
> **Response to `Reviewer Reply to Q2`**
>
> > Q2.1 Point-FEMAE vs. Point-PQAE
>
> The most significant concern is that Point-FEMAE adopts a different fine-tuning architecture compared to ours and Point-MAE, which **requires 3x the fine-tuning time cost.** **This potentially inflates the performance of fine-tuning tasks when training from scratch.**
>
> **Thus, this cannot be considered a fair comparison.** In contrast, we do not modify the encoder and add only a single layer of the PQ module between the encoder and decoder. The parameter count is 29.6M versus 29.0M for Point-MAE. **We maintain the same architecture as Point-MAE during fine-tuning.**
>
> Our Point-PQAE significantly outperforms the most relevant baseline (Point-MAE) and surpasses Point-FEMAE on some metrics (see 'Quick Response to Q2').
>
>
> > Q2.2 However, in this particular split, the proposed method demonstrates only limited improvement. In ReCon, even without foundation models, the performance still reaches 89.7.
>
> Please note that for the 89.73 in ReCon (single-modal), it uses the labels of the point cloud to implement MAE+contrastive (see lines 309-319 in [ReCon code](https://github.com/qizekun/ReCon/blob/main/models/ReCon.py), such as the code `if text[i] == text[j]:`), which is not explicitly illustrated in the original paper. **It is debatable whether using labeled data qualifies as an SSL method.**
>
> Additionally, the MAE+contrastive approach achieves 89.73 with more trainable parameters, longer training time, and by combining the advantages of contrastive and generative paradigms. In contrast, our Point-PQAE is a purely generative paradigm.
>
>
> > Q2.3 when considering Point-PQAE (APE), which combines components 1) and 3), the addition of multi-view reconstruction only results in limited performance gains, achieving 94.1, 92.1, and 88.2.
>
> We have addressed this in Q1.2 and Q1.3. **The setting in Point-PQAE (APE) combines components 1) and 3), but some are insufficiently decoupled.** Sufficient decoupling includes min-max normalization + Rotation.
>
> After cropping, it is crucial to recenter the instance and rescale it to a normalized size. Therefore, it is expected to observe lower performance (94.1, 92.1, and 88.2) without min-max normalization (deterministic Translate + Scale). With min-max normalization + Rotation, the performance improves to 95.0, 93.6, and 89.6.
>
> These results demonstrate the significant effectiveness of our multi-view approach over the single-view approach (see Line 4 or 5 vs. Line 1).
>
> ### Keep in mind that our RPE is essentially a normal PE with view-level information. It just happens to share the name accidentally with existing RPEs, so the comparison between Line 5 and Line 1 is valid.
>
> > Q2.4 If the author directly adds component 3) to Point-MAE in ReCon, what are the resulting metrics?
>
> Why do you believe that the position query module is applicable to a self-reconstruction framework and can directly increase the performance? We would be happy to hear your detailed opinion on this, as we have clarified this point multiple times before:
>
> We see it as a method to model the relationship between two views. Within the same view, however, self-attention in the decoder is sufficient. Simply adding a cross-attention layer (PQ module) between the encoder and decoder in ReCon for self-reconstruction **will not improve performance without doubt.** It's just can be seen an module redundant implementation of current ReCon.
>
> > Q2.5 The main improvement appears to be primarily introduced by the RPE.
>
> The main improvement comes from the multi-view approach, as shown in Line 4 and Line 5 versus Line 1. The reason we do not directly compare Line 4 or Line 5 to Point-MAE (rotation) is explained in the response to Q1.2:
>
> - **It is incorrect to solely compare the performance of each ablation to Point-MAE** because we use overlapping parts, whereas Point-MAE uses non-overlapping masked and visible parts.
>     - In the 2D domain, this distinction also exists between MAE and CropMAE. CropMAE uses overlapping views, while MAE uses non-overlapping masked and visible parts.
>     - As a result, reconstructing part 1 from part 2 is relatively trivial in the `single-view` setting. **It is even more trivial than in Point-MAE, which reconstructs non-overlapping parts.** This difference makes the starting points of these two paradigms fundamentally different.
>     - **It is more appropriate to compare Point-PQAE (APE) Line 1 as the vanilla baseline.**
>
>
> > Q2.6 I believe that adding any state-of-the-art learnable positional embedding would lead to performance improvements, regardless of whether RPE is used.
>
> ### Keep in mind that our RPE is exactly normal PE with view level information. Please see the answer to Q3.1.

---

> ### Author Response · Authors · 2024-11-23
> **Response to `Reviewer Reply to Q3`**
>
> > Q3.1 My point about "directly using" RPE is whether any specific RPE structure was uniquely designed for this paper.
>
> Yes.
>
> As you mentioned, `Moreover, to my knowledge, RPE is not specifically designed for the multi-view scenario`, we have clarified many times that our RPE **only shares the same name with existing RPE by coincidence which is designed for our cross-view method.** However, it seems you have not re-examined the form of our RPE compared to existing RPEs. For clarity, we provide the calculation below:
>
>
> For view 1 patch centers $C_i^1$, and view center $V_1$, view 2 patch centers $C_i^2$, and view center $V_2$:
>
>
> 1. PE in Point-MAE:
>
>     $PE_i^1 = \text{PE module} (C_i^1)$
>
> 2. (R)PE in Point-PQAE:
>
>     $PE_i^1 = \text{PE module} (V_2-V_1, C_i^1)$ for reconstructing view 1. It contains $V_2 - V_1$, thus we call it `Relative` PE.
>
> 3. RPE that you mean:
>
>     $PE_{ij}^1 = \text{Lernable modules} (C_i^1, C_j^1)$
>
> **2. and 3. are fundamentally different.**
>
> Our approach uses a normal PE form with relative view center information incoporated addtionally, which is why we call it Relative Positional Embedding (RPE). The name similarity to existing RPEs is purely coincidental.
>
> **If the term "RPE" is too misleading, we are willing to renaming it as RVPE (Relative View Positional Embedding).**
>
> In summary, our RVPE is uniquely designed for our multi-view task, while existing RPEs are not intended for multi-view scenarios.
>
> > Q3.2 Given that the main improvement in this paper seems to come from the use of RPE, it weakens the overall motivation for emphasizing multi-view reconstruction.
>
> We do not claim that the main improvement comes from RVPE; rather, RVPE simply fits our framework. Please refer to the answer to Q2.5. Your method of performance comparison is incorrect. RVPE is simply a mechanism to handle different coordinate systems and is only one part of our proposed Point-PQAE. **The main improvement stems from the cross-reconstruction paradigm, as opposed to the self-reconstruction paradigm.** This is demonstrated by comparing Line 4/Line 5 to Line 1. **All three components in our framework are designed to achieve this goal.**
>
> > Q3.3 Regarding performance gains, as I previously mentioned, the most critical benchmark is the PB-T50-RS split, where the observed improvement remains limited.
>
> **Please refer to the answers to Q1.2 and Q2.5.** Your method of comparison is somewhat flawed because you incorrectly assume that Translate and Scale are random augmentations, similar to Rotation, which we have already clarified is not the case. Moreover, **it is incorrect to directly compare the performance of each ablation to Point-MAE.** The effectiveness of our framework and the multi-view learning paradigm should be evaluated through Line 4 and Line 5 compared to Line 1. The observed performance gain is meaningful.
>
> ### **We are happy to address any concerns, but we feel deeply disappointed that our paper does not seem to have been read carefully. Despite repeated clarifications on the same question, significant misunderstandings persist.**
>
> ### **While we remain open to addressing your new concerns, we hope this rebuttal will clarify the fundamental misunderstandings that have persisted from the beginning.**

---

> > ### Comment · Reviewer_nNPL · 2024-11-23
> >
> > Thank you to the author for the correction. I apologize for my misunderstanding regarding RPE. However, I would like to clarify the results for the ReCon single modality setting: the scores of 94.15, 93.12, and 89.73 were not obtained using the text modality. The results of 94.66, 93.29, and 90.32 were achieved through contrastive learning without a foundation model, while the scores of 95.18, 93.63, and 90.63 were obtained using a foundation model.
> >
> > Currently, my primary concern lies with the results and the underlying motivation. In my view, point cloud multi-view reconstruction is a highly promising direction. However, in this context, referring to utilizing one crop part to reconstruct another as "multi-view" does not seem appropriate. Have the authors considered applying this method to the ScanNet dataset, which contains authentic multi-view point clouds with overlapping regions? I believe this approach would be far more meaningful for 3D scene understanding and is also compatible with the current framework, as modifying from ShapeNet to ScanNet would require minimal adjustments.
> >
> > If the authors explore this direction, I believe it would add significant value to the study. At present, the current object cropping approach doesn't appear particularly compelling. If the authors could conduct these experiments, provide visualizations, and release the code, I would be willing to raise my score. For now, I intend to adjust my score to borderline but a little bit leanto reject this paper, as I feel that this does not represent true multi-view reconstruction and the results are not particularly promising.  But I am ok with this paper getting accepted.

---

> > > ### Comment · Reviewer_nNPL · 2024-11-23
> > >
> > > For the multi-view setting in ScanNet or SUN-RGBD, PointContrast [1] can serve as a valuable reference. It generates paired point clouds and images, providing multi-view image and point cloud pairs within these datasets. I encourage the authors to explore 3D multi-view reconstruction or even multi-modal (2D/3D) multi-view reconstruction. I believe this direction would make for a highly compelling and interesting topic.
> > >
> > > [1] PointContrast: Unsupervised Pre-training for 3D Point Cloud Understanding

---

> ### Author Response · Authors · 2024-11-25
>
> Thank you for your prompt response and active engagement in the discussion. We are glad that the previous misunderstandings have been largely clarified through the rebuttal, and we appreciate your thoughtful reconsideration.
>
> We also thank you for clarifying the performance of ReCon. Now we have a clear understanding of it.
>
> As for your current primary concern:
>
> > Q1.1 However, in this context, referring to utilizing one crop part to reconstruct another as "multi-view" does not seem appropriate.
>
> It is more appropriate to define **different views** as two instances with a dynamic relative relationship, while **different parts** are two cropped sections of the same point cloud with a fixed relative relationship. This distinction has been validated through our experiments, where we compared the performance difference between using `parts` and `views`.
>
> **We argue that scene datasets like ScanNet do not inherently represent multi-view learning.** That is, `scene`$\neq$`multi-view`; instead, `multi-view` is determined by the dynamic relative relationships. **It is therefore inappropriate to claim that there is no `multi-view` learning at the object level.** At the object level, cropped parts can also be highly meaningful. For example, a cropped plane could represent a tail, a wing, or a combination of both. **By varying the relative relationships between these meaningful parts**, the reconstruction task becomes significantly more challenging than simple self-reconstruction, leading to substantial performance gains.
>
> In summary, our proposed Point-PQAE is the first method to explore cross-view reconstruction within the 3D SSL domain. **By leveraging the power of multi-view learning, we bring a novel perspective to the field and achieve significant performance improvements.**
>
> > Q1.2 Have the authors considered applying this method to the ScanNet dataset, which contains authentic multi-view point clouds with overlapping regions
>
> Thank you for your thoughtful question. **We claim that our Point-PQAE is a significant contribution to the 3D SSL domain, being the first method to achieve cross-view reconstruction.** This innovation has the potential to inspire advancements in other 3D tasks, including scene-level pre-training on datasets like ScanNet.
>
> As you pointed out, `I believe this approach would be far more meaningful for 3D scene understanding and is also compatible with the current framework, as modifying from ShapeNet to ScanNet would require minimal adjustments.` We agree that insights from our object-level cross-view reconstruction can undoubtedly extend to more complex 3D tasks, including scene-level learning.
>
> However, we emphasize that our method represents the first meaningful and successful attempt at multi-view reconstruction in self-supervised learning for 3D point clouds. **This achievement is sufficient to establish the novelty and contribution of our approach as a new methodology.**
>
> Regarding pre-training on ScanNet, we appreciate your suggestion and acknowledge this as a promising direction for exploring scene-level cross-view reconstruction. This is a derivative topic from Point-PQAE, and we plan to investigate it in future work.
>
> We encourage you to reconsider the significance of our object-level multi-view learning and recognize the contribution our method brings to the field.

---

> ### Author Response · Authors · 2024-11-25
> **Confirm whether the doubt over the effectiveness of Point-PQAE has been resolved**
>
> Dear Reviewer nNPL,
>
> First, we would like to thank you again for your active engagement in the discussion. While some misunderstandings regarding the RPE formulation and the experimental settings in the ablation study led to questions about the effectiveness and novelty of our method, **we are grateful to have addressed these concerns through the rebuttal.** Thank you for reading it in detail.
>
> Now, **considering that reviewers Gtpy and eEi5 agree with your earlier concerns about the effectiveness of our method, which we believe have already been addressed**, and have maintained a negative stance, we kindly ask for a clear response on whether these concerns have indeed been resolved.
>
> Specifically, the ablation study demonstrating the effectiveness of our approach focuses on cross-view reconstruction versus single-view reconstruction, as shown below:
>
> |Method|Multi-view|min-max norm part 1 (Decentralization)|min-max norm part 2 (Scale to [-1, 1])|Rotation|OBJ-BG|OBJ-ONLY|PB-T50-RS|
> -|-|-|-|-|-|-|-
> |Point-PQAE (APE)|✗|✗|✗|✗|92.3|91.0|87.7
> |Point-PQAE (APE)|✓|✗|✓|✗|94.1|92.1|88.2
> |Point-PQAE (APE)|✓|✗|✗|✓|93.6|92.4|88.8
> |Point-PQAE (APE)|✓|✗|✓|✓|93.8|92.8|88.8
> |Point-PQAE (RPE)|✓|✓|✓|✓|95.0|93.6|89.6
>
> The results, particularly the comparison between line 5 and line 1, clearly demonstrate the effectiveness of our proposed cross-view learning approach. For a more detailed analysis, we invite you to review the explanation provided in Q1.2 of the "Response to \`Reviewer Reply to Q1\` part 1."
>
> **Your response is highly valuable and will play a crucial role in shaping the final evaluation of our work.**

---

> > ### Author Response · Authors · 2024-11-30
> > **Look forward to your further reply**
> >
> > Dear Reviewer nNPL,
> >
> > Thank you once again for your valuable and insightful suggestions. We will certainly consider exploring these ideas in future work.
> >
> > **Our proposed Point-PQAE method leverages cross-view learning at the object level, which introduces a novel perspective to the field and leads to significant performance improvements.** We believe this approach has the potential to inspire future research, including, as you suggested, the application of cross-view learning at the scene level.
> >
> > Considering that reviewers Gtpy and eEi5 agree with your earlier concerns about the effectiveness of our method, which we believe have already been addressed, **we would kindly like to know if the concern regarding the greater effectiveness of the multi-view learning approach we proposed, compared to the single-view case, has been addressed.**
> >
> > **Apologies for sending this message again, but we believe this issue is crucial in shaping the final evaluation of our work.** We also apologize for not adequately explaining certain aspects of our experimental setup in the initial rebuttal, which may have led to your confusion.
> >
> > **Your response is highly valuable.**
> >
> > Kind regards,
> > The Authors

---

> ### Comment · Reviewer_nNPL · 2024-12-02
>
> Thank you to the authors for the detailed response. I appreciate the ablation studies provided, which demonstrate the effectiveness of each component. However, the improvement over Point-MAE (ReCon version with augmentation) does not appear to be substantial, and this work falls short of achieving state-of-the-art (SOTA) performance.
>
> Another concern I have is regarding the definition of "multi-view." In this paper, the concept of reconstructing one part of an object from another is more akin to completion [1] rather than multi-view learning. Typically, multi-view is associated with utilizing 2D information to describe a 3D object or scene, as seen in previous works [2, 3, 4, 5, 6, 7]. In earlier studies on 3D object learning, multi-view (e.g., [2, 3, 4]) generally refers to leveraging multiple views of images to enhance the understanding of a 3D object. Similarly, in 3D scene understanding, multi-view approaches [5, 6, 7] utilize the overlapping regions of point clouds from different 2D perspectives. Therefore, I do not believe this paper presents a "multi-view learning approach." Rather, I consider it a new data augmentation strategy. As I highlighted earlier, the proposed method could be more aligned with multi-view learning if it were applied at the scene level—reconstructing one view from another at the scene level would make it truly representative of a multi-view learning approach.
>
> In the previous rebuttal, the authors argued that scene datasets like ScanNet do not inherently represent multi-view learning. I disagree with this point, as many prior works have effectively utilized multi-view information [5, 6, 7] to enhance scene understanding tasks.
>
> Considering this, I will keep my score.
>
>
> [1] PoinTr: Diverse Point Cloud Completion with Geometry-Aware Transformers
>
> [2] Learning Relationships for Multi-View 3D Object Recognition
>
> [3] Voint Cloud: Multi-View Point Cloud Representation for 3D Understanding
>
> [4] Multi-view Convolutional Neural Networks for 3D Shape Recognition
>
> [5] Multi-View Representation is What You Need for Point-Cloud Pre-Training
>
> [6] Learning Multi-View Aggregation In the Wild for Large-Scale 3D Semantic Segmentation
>
> [7]3DMV: Joint 3D-Multi-View Prediction for 3D Semantic Scene Segmentation

---

> > ### Author Response · Authors · 2024-12-02
> > **Part 1 (effectiveness concern)**
> >
> > Thank you for your response.
> >
> > At first we thank you for reading our rebuttal and agree with the performance of the ablations (check the 'Global official comment
> > ' for ablations for our Point-PQAE (cross-view case reconstruction) vs. single-view case reconstruction).
> >
> > **The effectiveness of our Point-PQAE is demonstrated through two key aspects: the ablation studies and the performance improvement** over the baseline. The issues regarding the effectiveness of the ablation studies are addressed, and the performance improvements can be clearly illustrated with quantitative results:
> >
> > > Q1.1 However, the improvement over Point-MAE (ReCon version with augmentation) does not appear to be substantial
> >
> > **It's indeed a substantial improvement of our Point-PQAE over Point-MAE (ReCon version with augmentation).**
> >
> > This question was answered in a previous rebuttal in the comment to you titled 'Quick response to Q2' (The answer to Q2.2), which can be summarized as follows:
> >
> > ScanObjectNN (FULL, Mlp-Linear, Mlp-3)|OBJ-BG|OBJ-ONLY|PB-T50-RS
> > -|-|-|-
> > Point-MAE (ReCon version with augmentation)|92.6|91.9|88.4|
> > Point-PQAE (ours)|**95.0**|**93.6**|**89.6**
> > Point-MAE (ReCon version with augmentation)|82.8±0.3|83.2±0.2|74.1±0.2
> > Point-PQAE (ours)|**89.3±0.3**|**90.2±0.4**|**80.8±0.1**
> > Point-MAE (ReCon version with augmentation)|85.8±0.3|85.5±0.2|80.4±0.2
> > Point-PQAE (ours)|**90.7±0.2**|**90.9±0.2**|**83.3±0.1**
> >
> > ModelNet | 1k | 8k | 5w10s | 5w20s | 10w10s | 10w20s
> > -|-|-|-|-|-|-
> > Point-MAE (ReCon version with augmentation)|93.8|94.0|96.4±2.8|97.8±2.0|92.5±4.4|95.2±3.9
> > Point-PQAE (ours)|**94.0**|**94.3**|**96.9±3.2**| **98.9±1.0**| **94.1±4.2**| **96.3±2.7**
> >
> > **In most of the benchmarks, our Point-PQAE outperforms the Point-MAE (ReCon version with augmentation) significantly.**
> >
> > ### So could you please tell me the reason you think Point-PQAE doesn't substantially outperform the baseline?
> >
> > *Note: Simply adding random crop to Point-MAE doesn't improve the performance (Check table 4 (c) in the main paper).*
> >
> > > Q1.2 This work falls short of achieving state-of-the-art (SOTA) performance.
> >
> > First, we (single-modal) achieve competitive performance compared to ReCon (cross-modal SOTA), achieving SOTA in some benchmarks. Please see the detailed comparison in 'Quick response to Q2' (The answer to Q2.2).
> >
> > We have indeed stated why Point-PQAE doesn't achieve SOTA in all benchmarks in a previous rebuttal and in Appendix Section E: Limitations and Future Works:
> > > **Compared to the well-studied self-reconstruction, cross-reconstruction remains significantly under-explored.** As the initial venture into cross-reconstruction, our Point-PQAE opens a new avenue for advancement in point cloud SSL. However, the model employs a vanilla transformer architecture and is constrained to single-modality knowledge currently. This architecture may not be optimally suited for cross-reconstruction tasks and further exploration towards incorporating cross-modal knowledge will possibly enhance the performance.

---

> > ### Author Response · Authors · 2024-12-02
> > **Part 2 (multi-view concern)**
> >
> > > Q2.1 Definition of the multi-view in our paper.
> >
> > Regarding your statement, "In this paper, the concept of reconstructing one part of an object from another is more akin to completion [1] rather than multi-view learning," **we have repeatedly emphasized the importance of distinguishing `parts` and `views`:**
> >
> >
> > - **Relative relationships between the cropped `parts` remain fixed.** For example, in Point-MAE (block mask), the visible and masked blocks are augmented as a whole, keeping their relative relationships fixed.
> > - **By performing view decoupling, the relative relationships between parts become more diverse, and we define these as `views`.**
> >
> > The experimental results clearly demonstrate the importance of distinguishing between `parts` and `views`. Cross-`view` reconstruction significantly outperforms cross-`part` reconstruction (see the 'Global official comment,' line 3 vs. line 1):
> >
> > **Note that line 1 is similar to Point-MAE (block mask) and PoinTr [1]. They are all cross-`part` reconstruction.**
> >
> > Cross-`view` reconstruction, using our pipeline where `views` are obtained through view decoupling, enables more challenging and informative pre-training, and easily outperforms cross-`part` reconstruction.
> >
> > > Q2.2 Definition of the multi-view in the papers that you mentioned.
> >
> > Thank you for mentioning additional papers that help us better position our method.
> >
> > The multi-view concepts in the works you mentioned [2][3][4][5][6][7] share the same definition: different views are 2D projections/renderings of 3D objects/scenes from different viewpoints.
> >
> >
> > **Relation to our multi-view:**
> >
> > - In our approach, **view 1 and view 2 can also be thought of as being obtained from different viewpoints.** Recall that after independently cropping and applying min-max normalization, view 1 and view 2 are augmented (rotation) in isolation. This is equivalent to maintaining the world coordinates and rotating the 'eye'/'camera' to observe the point cloud. This is similar to the different viewpoints in the papers you mentioned.
> >
> >
> > Difference:
> >
> > - **We retain multi-view in the 3D form**, whereas the mentioned works project/render data into 2D.
> > - **We use isolated cropped point clouds to obtain multi-view**, while works [2][3][4] use multi-viewpoint 2D projections/renderings from the same point clouds.
> > - **We obtain multi-view by using cropped parts, with the relative relationships varied randomly.** The independently performed augmentations (i.e. eye position changes) are not provided to the model. The works you mentioned do not focus on the relative relationships but instead use them to aggregate 3D features.
> >
> > From this perspective, and based on our answer to Q2.1, we believe that our method represents a novel paradigm (**cross/multi-view reconstruction**) for point cloud self-supervised learning, **rather than just a new data augmentation strategy as you suggested.**
> >
> > > Q2.3 As I highlighted earlier, the proposed method could be more aligned with multi-view learning if it were applied at the scene level—reconstructing one view from another at the scene level would make it truly representative of a multi-view learning approach.
> >
> > We do not find the explicit logic behind this opinion.
> >
> > We agree with you that scene-level multi-view cross-reconstruction learning is also important. **However, as seen in the works you mentioned, multi-view learning exists both at the object level [2][3][4] and the scene level [5][6][7].** So, why would simply transferring from object-level to scene-level make a significant difference? Would it make cross-reconstruction at the object level meaningless, but valid at the scene level as true multi-view learning?
> >
> > Thus, cross-view reconstruction is meaningful at both the object and scene levels. **Our proposed Point-PQAE method** leverages multi-view cross-reconstruction at the object level, leading to significant performance improvements. We believe this approach has the potential to inspire future research, including, as you suggested, the application of cross-view learning at the scene level.
> >
> > > Q3 Scene datasets represent multi-view learning.
> >
> > **It is more appropriate to say that multi-view learning exists at the scene level**, rather than saying that scene datasets *represent* (or are equal to) multi-view learning.
> >
> > **Multi-view learning also exists at the object level**, as seen in works [2][3][4], as well as in our cross-view reconstruction (see the answer to Q2.3).
> >
> > Reference
> >
> > [1] PoinTr: Diverse Point Cloud Completion with Geometry-Aware Transformers
> >
> > [2] Learning Relationships for Multi-View 3D Object Recognition
> >
> > [3] Voint Cloud: Multi-View Point Cloud Representation for 3D Understanding
> >
> > [4] Multi-view Convolutional Neural Networks for 3D Shape Recognition
> >
> > [5] Multi-View Representation is What You Need for Point-Cloud Pre-Training
> >
> > [6] Learning Multi-View Aggregation In the Wild for Large-Scale 3D Semantic Segmentation
> >
> > [7]3DMV: Joint 3D-Multi-View Prediction for 3D Semantic Scene Segmentation

---

> ### Comment · Reviewer_nNPL · 2024-12-02
>
> **For the Results:**
>
> The claimed improvements over Point-MAE (2.4, 1.7, 1.2) can not be regarded as significance. One concurrent work, PCP-MAE [1], which also uses only point clouds, achieved 95.52, 94.32, and 90.35 accuracy with only 22.1M parameters. Interestingly, PCP-MAE still compared with Point-FEMAE and ReCon and admitted that they only achieved state-of-the-art (SOTA) results in certain part splits. Given that PCP-MAE is a concurrent work, it may not be mandatory for comparison. However, considering that the metrics presented in this paper are consistently lower than those of PCP-MAE, yet the authors still insist on claiming SOTA, it seems necessary to mention this paper as an example to illustrate that the work does not meet SOTA benchmarks.
>
> **Regarding Multi-View Learning:**
>
> The term "multi-view" traditionally refers to the collection of multiple 2D views or perspectives of a 3D object or scene. However, it is unclear how this concept is represented in this paper. The authors seem to regard different augmentation parts as different "multi-views." Could the authors reference a related work on 3D object-level learning that uses this definition of "multi-view"? If not, it seems the authors may have introduced their own definition of multi-view learning for point clouds. This raises questions about the validity of their claim to learning multi-view representations for point clouds.
>
> For scene-level 3D learning methods [2, 3, 4], multi-view typically involves multiple perspectives for 3D scene understanding, which is a pre-defined task. It is important to note that defining multi-view learning should align with previous works, rather than introducing new interpretations without sufficient justification.
>
> **Object-Level vs. Scene-Level Learning:**
>
> The suggestion to transition from object-level to scene-level representations implies a significant distinction. Reconstructing from one part of an object to another via augmentation does not adequately qualify as multi-view learning at the object level. In contrast, in scene-level 3D learning, prior work often uses multiple perspectives of a scene for better 3D scene understanding. Hence, reconstructing a point cloud from one perspective to another can justifiably be regarded as multi-view at the scene level. The current work's approach to object-level reconstruction cannot make me believe it can benefit multi-view representation learning.
>
> As I mentioned before, I will keep my original score, but I am okay with this paper being accepted.
>
> [1] PCP-MAE: Learning to Predict Centers for Point Masked Autoencoders
>
> [2] Multi-View Representation is What You Need for Point-Cloud Pre-Training
>
> [3] Learning Multi-View Aggregation In the Wild for Large-Scale 3D Semantic Segmentation
>
> [4] 3DMV: Joint 3D-Multi-View Prediction for 3D Semantic Scene Segmentation

---

> ### Author Response · Authors · 2024-12-03
> **Response 1 performance**
>
> Thank you for your response and for being okay with the acceptance of our work. The concerns of performance improvement significance and the definition of 'two-view'/'multi-view' can be addressed easily:
>
> > Q1. The concurrent work and the claimed improvements over Point-MAE (2.4, 1.7, 1.2) can not be regarded as significance.
>
> We understand that our views on the definition of "significance" may differ. We present the improvement comparison (over the Point-MAE baseline) between ReCon and our approach:
>
> Method|Param(M)|Time(h)|BG(F)|OY(F)|RS(F)|BG(L)|OY(L)|RS(L)|BG(3)|OY(3)|RS(3)|MN1k|MN8k|5w10s|5w20s|10w10s|10w20s|
> -|-|-|-|-|-|-|-|-|-|-|-|-|-|-|-|-|-
> Ours|**29.5**|**19.6**|2.4|**1.7**|1.2|6.5|**7.0**|6.7|**4.9**|**5.4**|2.9|0.2|0.3|0.5|**1.1**|**1.6**|**1.1**
> ReCon (Cross-modal SOTA)|140.9|44.4|**2.6**|**1.7**|**2.2**|**6.7**|6.5|**7.3**|4.8|5.2|**3.4**|**0.7**|**0.7**|**0.9**|**1.1**|0.8|0.6
>
> *BG: OBJ-BG, OY: OBJ-ONLY, RS: PB-T50-RS
>
> *F: FULL, L: Mlp-Linear, 3: Mlp-3
>
> If ReCon (cross-modal SOTA) is considered a significant improvement over Point-MAE, we believe the same can be said for our Point-PQAE (single-modal).
>
> **Thank you for mentioning the concurrent work PCP-MAE.** While our Point-PQAE primarily compares with the most relevant baseline, Point-MAE (rotation), we will revise our SOTA claim to "outperform Point-MAE significantly" for the sake of rigor (this change will be reflected in the next version).
>
> What we want to emphasize is that '**compared to the well-studied self-reconstruction, cross-reconstruction remains significantly under-explored.**' For instance, developing PCP-PQAE (by adopting ideas from PCP-MAE) could involve predicting the centers of view 2 from view 1, potentially leading to a significant performance boost. We will include a discussion of the advanced MAE-based methods (Point-FEMAE and PCP-MAE) in the appendix.

---

> ### Author Response · Authors · 2024-12-03
> **Response 2 related works that use two-view/multi-view**
>
> > Q2 Could the authors reference a related work on 3D object-level learning that uses this definition of "multi-view"? If not, it seems the authors may have introduced their own definition of multi-view learning for point clouds.
>
> We have already sufficiently addressed this point in the main paper (lines 16-20, 81-100, etc.) **by stressing that we use two-view cross-reconstruction due to its greater diversity and citing related works.** Note that we use the term "two-view" rather than "multi-view" in our paper. **The definition of "two-view" we use is widely adopted across various domains, with several examples:**
>
> - In 2D contrastive methods, such as SimCLR [1], BYOL [2], and DINO [3], the term "two views" refers to isolated augmented pairs, such as "two correlated views," "different augmented views of the same image," or "a pair of views."
> - In 2D cross-reconstruction methods like CropMAE [4], "two views" also refers to isolated augmented pairs.
> - In 3D contrastive methods, such as PointContrast [5], it is stated: "From x, generate two views x_1 and x_2," and "We then sample two random geometric transformations T1 and T2 to further transform the point clouds into two views. The transformation makes the pretext task challenging, as the network must learn equivariance to the geometric transformations imposed. In this work, we mainly consider rigid transformations, including rotation, translation, and scaling." (3D data from different viewpoints can be viewed as equivalent to different rotations to some extent.)
>
> The term "two-view" is used broadly across these works. It is acceptable to refer to "two-view" as "multi-view" in some contexts, but not the other way around. Thus, in some parts of the rebuttal, we use the term "multi-view." **We adopt this definition from other domains/methods naturally, rather than defining it ourselves, and we refer to the isolated augmented point clouds as "two-view"** All in all, "two-view" should not be confined to just two viewpoints, as you might interpret it.
>
> ### Therefore, we do not introduce new interpretations without sufficient justification but rather align with established definitions from related fields.
>
> > Q3. Object-Level vs. Scene-Level Learning:
>
> In the context of related two-view works [1][2][3][4][5] (which can certainly be referred to as two-view learning), **our Point-PQAE reconstructs one view (an isolated augmented part) from another view of an object, and this can also be classified as two-view (or multi-view) learning without question.**
>
> If you prefer that we avoid mixing the terms 'two-view' and 'multi-view,' we can certainly standardize our terminology to only use 'two-view,' although we believe that 'two-view' can also be considered a form of 'multi-view.'
>
> We agree that using multiple perspectives of a scene is crucial for better 3D scene understanding. **However, the works you referenced [6][7][8] also demonstrate that using multiple perspectives for objects is equally meaningful.** Therefore, there is no strong rationale for undermining the significance of object-level multi-view learning in favor of scene-level learning.
>
> Reference
>
> [1] A Simple Framework for Contrastive Learning of Visual Representations
>
> [2] Bootstrap Your Own Latent A New Approach to Self-Supervised Learning
>
> [3] Emerging Properties in Self-Supervised Vision Transformers
>
> [4] Efficient Image Pre-Training with Siamese Cropped Masked Autoencoders
>
> [5] PointContrast：Unsupervised Pre-training for 3D point cloud understanding
>
> [6] Learning Relationships for Multi-View 3D Object Recognition
>
> [7] Voint Cloud: Multi-View Point Cloud Representation for 3D Understanding
>
> [8] Multi-view Convolutional Neural Networks for 3D Shape Recognition

---

> ### Comment · Reviewer_nNPL · 2024-12-03
>
> Thank you to the author for the detailed response. Referring to the augmented parts as "views" is common practice in the self-supervised learning domain. However, in the case of SimCLR, DINO, and PointContrast, none of these works explicitly claim to be multi-view learning and trained to learn **multi-view features**. Instead, what they focus on is learning **augmentation-invariant features**.
>
> The concept of multi-view learning is particularly important at the object level. However, as I mentioned earlier, the term "3D multi-view" traditionally refers to the **collection of multiple 2D views or perspectives of a 3D object or scene**. In the 3D object level, this involves capturing multiple 2D projections of a 3D object to represent its structure comprehensively [1, 2, 3]. Unfortunately, I find it difficult to align this work with that definition.
>
> Therefore, I believe this paper does not fall under the category of 3D multi-view learning. Instead, it utilizes a single view combined with block masking and augmentation techniques. As such, the motivation for this paper may need to be revised to accurately reflect its methodological focus.
>
> Regarding the results, as mentioned earlier, the most important setting is the fine-tuning of the PB-T50-RS part, as it presents the greatest challenge. However, the improvement is this setting is not good enough. In terms of the MLP-learner results, some papers even do not report them, as linear probing often fails to adequately reflect representation learning and has a large variance, particularly for smaller models.
>
> [1] Learning Relationships for Multi-View 3D Object Recognition
>
> [2] Voint Cloud: Multi-View Point Cloud Representation for 3D Understanding
>
> [3] Multi-view Convolutional Neural Networks for 3D Shape Recognition

---

> ### Author Response · Authors · 2024-12-03
> **Quick response to nNPL**
>
> Thank you for your quick response and active discussion.
>
> > Q1. What they focus on is learning augmentation-invariant features.
>
> It's correct that these two-view contrastive methods learn augmentation-invariant features, **but it's also true that there exist greater diversity between the two views over single view which they rely on.** Without this diversity, contrastive learning would be meaningless when the same instance is input.
>
> **This exactly highlights the novelty of our work over these contrastive methods.** Given the advantage of two-view, we introduce it into a generative paradigm (Point-MAE) to enable more difficult and informative pre-training (our Point-PQAE).
>
> > Q2. Unfortunately, I find it difficult to align this work with that definition (multi-view(point) learning paradigm).
>
> Yes, our work doesn't aim to align with multi-view(point) learning methods that you mentioned. We have illustrated the definition of two-view in our method and discuss the difference to multi-view(point) in previous rebuttal.
>
> In the context of related two-view works [1][2][3][4][5] (which can certainly be referred to as two-view learning), our Point-PQAE reconstructs one view (an isolated augmented part) from another view of an object, bringing the great diversity in two-view to 3D SSL and leading improved performance.
>
> > Q3. Therefore, I believe this paper does not fall under the category of 3D multi-view learning. Instead, it utilizes a single view combined with block masking and augmentation techniques.
>
> Thank you for this question. Our Point-PQAE does not fall under the category of 3D multi-view(point) learning, but it definitely falls within the two-view learning paradigm (e.g., SimCLR, DINO, BYOL, CropMAE, PointContrast), where related works exist and have been shown to be effective.
>
> We do not consider the classic work DINO [3] as merely a technique that combines a single view with crop and augmentation techniques. This is also the case for our Point-PQAE.
>
> > Q4. Performance
>
> We have provided sufficient experimental results. If ReCon (cross-modal SOTA) is considered a significant improvement over Point-MAE, we believe the same can be said for our Point-PQAE (single-modal).
>
> Additionally, linear probing is a way to assess the quality of the pre-trained representations directly. Variations across different runs do exist, so we report both the average and variance. We observe significant performance improvement:
>
> ScanObjectNN (Mlp-Linear, Mlp3)|OBJ-BG|OBJ-ONLY|PB-T50-RS|
> |-|-|-|-
> Point-MAE|82.8±0.3|83.2±0.2|74.1±0.2|
> Point-PQAE (ours)|89.3±0.3|90.2±0.4|80.8±0.1|
> Point-MAE|85.8±0.3|85.5±0.2|80.4±0.2|
> Point-PQAE (ours)|90.7±0.2|90.9±0.2|83.3±0.1|
>
>
> [1] A Simple Framework for Contrastive Learning of Visual Representations
>
> [2] Bootstrap Your Own Latent A New Approach to Self-Supervised Learning
>
> [3] Emerging Properties in Self-Supervised Vision Transformers
>
> [4] Efficient Image Pre-Training with Siamese Cropped Masked Autoencoders
>
> [5] PointContrast：Unsupervised Pre-training for 3D point cloud understanding

---

### Official Review · Reviewer_qqpL · 2024-11-04

**Soundness:** 3
**Presentation:** 4
**Contribution:** 3
**Rating:** 8
**Confidence:** 5

**Summary:**

This paper proposes Point-PQAE, a novel SSL framework for 3D point cloud data. Unlike traditional single-view reconstruction, Point-PQAE uses cross-reconstruction, generating two decoupled views of the same point cloud and reconstructing one from the other. This approach leverages relative positional encoding and a position-aware query module, leading to enhanced performance on 3D object classification and segmentation tasks. The framework demonstrates significant improvements over previous SSL methods.

**Strengths:**

1. The idea of using cross-reconstruction is nice.
2. The performance gain is meaningful.
3. Overall presentation/writing is very good.

**Weaknesses:**

I think this paper has no major flaws.
Please see the questions.

**Questions:**

1. Can this approach (i.e., cross-recon) is applicable to image or text domain? Or are there similar works in those modalities?

2. I am curious about the impact of increasing the cross-reconstruction branch. For example, can the authors test triple-recon setting by partitioning the original point cloud into three different views?

3. Line 247, missed "." after RPE.

---

> ### Author Response · Authors · 2024-11-20
> **Response to Q1**
>
> Thank you for your time, detailed comments, and valuable suggestions. We are delighted that you recognize the writing quality, novelty, and great performance of our Point-PQAE. Here are our responses to the questions you raised:
>
> > Q1. Can this approach (i.e., cross-recon) be applicable to image or text domain? Are there similar works in those modalities?
>
> Yes, and yes. First, there are two essential components in the proposed cross-reconstruction method:
>
> 1. Two isolated/decoupled views, rather than two parts of the same instance that maintain a fixed relative relationship.
> 2. A model that achieves cross-view reconstruction using relative position information.
>
> **For the application of cross-reconstruction to the text domain:** The different 'views' can be obtained by randomly selecting two sentences from one article. Different paired sentences possess different relative relationships which meets the essential component 1 in the cross-reconstruction method. The relative positional information can be incorporated into cross-reconstruction. The specific methodology would need to be carefully designed. To the best of our knowledge, there are currently no similar works in the text domain.
>
> **For the application of cross-recon to the image domain:** By utilizing augmentations such as random crop, rotation, color distortion and etc, we can generate multiple views from a single image and the corresponding Relative Positional Embedding (RPE) can be incorporated for cross-reconstruction.  **There have been cross-reconstruction methods similar to our Point-PQAE proposed in the image domain, including SiamMAE [1] and CropMAE [2].** Instead of generating multiple views, SiamMAE utilize two different frames in the video and reconstruct the future frame from the past one. CropMAE alleviates the need for video datasets by using image augmentations to generate multiple views and reconstruct one from another. Both methods do not incorporate relative information into training but rely on non-fully masking to guide the cross-reconstruction between views. **Compared to them, the explicit RPE proposed by us could be more stable and have better explainability.**

---

> ### Author Response · Authors · 2024-11-20
> **Response to Q2-Q3**
>
> > Q2. The impact of increasing the cross-reconstruction branch. For example, can the authors test triple-recon setting by partitioning the original point cloud into three different views?
>
> Thank you for your helpful suggestion. We test the performance of our Point-PQAE with the triple-reconstruction setting by first independently generating three decoupled views and passing them separately through the encoder. We use paired cross-reconstruction and compute six RPEs for reconstruction. This results in six terms in the ${L}_{triple}$ loss, i.e.,
>
> $L_{triple} = L_{2→1}+L_{3→1}+L_{1→2}+L_{3→2}+L_{1→3}+L_{2→3}$
>
> Recall that for the double-recon (original Point-PQAE), the loss is (Eq. 11):
>
> $L_{double} = L_{2→1}+L_{1→2}$
>
> We pre-train the Point-PQAE with triple-recon and test it on the downstream ScanObjectNN classification task. The performance comparisons are shown as follows:
>
> Setting|pre-training time (h)|OBJ-BG|OBJ-ONLY|PB-T50-RS
> -|-|-|-|-
> **double-recon (ckpt-epoch-300)**|19.6 (1.00x)|95.0 (↑0.0)|93.6 (↑0.0)|89.6 (↑0.0)
> double-recon (ckpt-epoch-150)|9.8 (0.50x)|93.6 (↓1.4)|92.6 (↓1.0)|88.5 (↓1.1)
> double-recon (ckpt-epoch-100)|6.5 (0.33x)|92.4 (↓2.6)|91.9 (↓1.7)|87.8 (↓1.8)
> triple-recon (ckpt-epoch-300) |32.9 (1.68x)|94.8 (↓0.2)|93.6 (↑0.0)|89.5 (↓0.1)
> **triple-recon (ckpt-epoch-150)** |16.4 (0.84x)|94.5 (↓0.5)|93.6 (↑0.0)|89.4 (↓0.2)
> triple-recon (ckpt-epoch-100) |11.0 (0.56x)|93.8 (↓1.2)|92.9 (↓0.7)|88.8 (↓0.7)
>
> The pre-training time per epoch for double-recon is 234.9s (1.00x) and for triple-recon it is 394.8s (1.68x). Our analysis reveals several key advantages of the triple-recon setting:
>
> 1. **Much richer training paired samples with manageable computational costs.** When training for the same number of epochs, the number of training samples (paired data) for triple-recon is **three times** that of double-recon, as analyzed through the loss function. However, the training time does not increase linearly, i.e., 234.9s (1.00x) vs. 394.8s (1.68x). As for pre-training GPU memory usage, it is 19.26GB (1.00x) vs. 21.28GB (1.10x).
>
> 2. **Faster convergence and higher pre-training efficiency.** Pre-training Point-PQAE with the triple-recon setting may be a more efficient approach. From the results, we can see that triple-recon (epoch 150) already performs competitively with double-recon (epoch 300) while taking less time. Triple-recon has already converged after 150 epochs, which is earlier than double-recon due to the richer training samples.
>
> And the results also indicate an inspiring phenomenon: **After convergence, simply increasing the number of training epochs does not significantly improve performance.** The performance of triple-recon (ckpt-epoch-150) is already competitive with double-recon (ckpt-epoch-300). However, the improvement of triple-recon (ckpt-epoch-300) over ckpt-epoch-150 is marginal. Increasing the size of the pre-training dataset, rather than simply extending the training time, may help improve performance.
>
> **All in all, triple-recon is a promising pre-training acceleration technique that can achieve competitive performance with less training time than double-recon.** Thank you again for your insightful suggestion.
>
> > Q3. Line 247, missed "." after RPE.
>
> Thank you for your exceptionally thorough review. We have corrected the typo and uploaded the [revised PDF](https://openreview.net/pdf?id=jZASmAlxp2).
>
> Reference
>
> [1] Gupta A, Wu J, Deng J, et al. Siamese masked autoencoders[J]. NeurIPS 2023.
>
> [2] Eymaël A, Vandeghen R, Cioppa A, et al. Efficient Image Pre-Training with Siamese Cropped Masked Autoencoders[C]//ECCV 2024.

---

> ### Comment · Reviewer_qqpL · 2024-11-23
>
> I carefully read the rebuttals, and my concerns are properly addressed.
>
> However, I found some (conceptually) similar works in the imagery domain exist, as mentioned in the rebuttal of the authors, which weakens the key novelty of this paper.
>
> Therefore, I would like to lower my rating to 7, but the system currently does not allow it.
> So, I'll keep it 8, and I believe that AC would properly consider my stance.
>
> Although it seems that there remain some issues (tons of discussion between the authors and the reviewer nNPL), I still think this paper can bring meaningful contributions to the field of point cloud.

---

> > ### Author Response · Authors · 2024-11-24
> >
> > Thank you very much for your response and for recognizing our Point-PQAE.
> >
> > **We show the detailed comparison of our Point-PQAE over "some (conceptually) similar works in the imagery domain" including SiamMAE [1] and CropMAE [2] for clarity.** Hope this would be helpful for you to better comprehend our method:
> >
> > There are two essential components in our proposed cross-reconstruction framework:
> >
> > 1. Two isolated/decoupled views, rather than two parts of the same instance that maintain a fixed relative relationship.
> > 2. A model that achieves cross-view reconstruction using relative position information.
> >
> > **Our Point-PQAE pioneers the cross-reconstruction paradigm in 3D point cloud self-supervised learning (SSL)**, and to our knowledge, there are no similar previous methods in this domain. To better position our methodology, we compare it to similar SSL methods, including SiamMAE [1] and CropMAE [2], proposed in the image domain.
> >
> > **SiamMAE** operates on pairs of randomly sampled video frames and asymmetrically masks them, utilizing the past frame to predict the masked future frame. **CropMAE** relies on image augmentations to generate two views, using one to reconstruct the other.
> >
> > Relation to our Point-PQAE:
> >
> > 1. **All are cross-reconstruction methods.** Both SiamMAE and CropMAE have two essential components for cross-reconstruction-framework including multi-view (different frames sampled from one video for SiamMAE and isolated augmented images for CropMAE) and cross-view reconstruction model. They can be treated as cross-reconstruction methods, similar to our Point-PQAE.
> >
> > Difference:
> >
> > 1. **Different domain:** SiamMAE and CropMAE focus on the 2D SSL domain. Our Point-PQAE is the first method for cross-reconstruction in the 3D SSL domain.
> > 2. **Asymmetric/symmetric reconstruction:** SiamMAE uses the past to predict the future, which is asymmetric. CropMAE performs asymmetric reconstruction and doesn't explore siamese cross-reconstruction. In contrast, our Point-PQAE is inherently symmetric, and the siamese loss brings a performance gain.
> > 3. **No relative information utilized:** SiamMAE and CropMAE do not incorporate relative information into training but rely on non-fully masking to guide the cross-reconstruction. The RPE adopted by our Point-PQAE provides explicit guidance, making training more stable and improving explainability.
> > 4. **No tuned-needed mask ratio exists in our framework.** There is a hyperparameter mask ratio that needs to be tuned in both SiamMAE and CropMAE, but this is not the case in our Point-PQAE framework.
> >
> > Reference
> >
> > [1] Gupta A, Wu J, Deng J, et al. Siamese masked autoencoders[J]. NeurIPS 2023.
> >
> > [2] Eymaël A, Vandeghen R, Cioppa A, et al. Efficient Image Pre-Training with Siamese Cropped Masked Autoencoders[C]//ECCV 2024.

---

### Official Review · Reviewer_eEi5 · 2024-11-05

**Soundness:** 3
**Presentation:** 4
**Contribution:** 3
**Rating:** 5
**Confidence:** 3

**Summary:**

The paper proposed a novel cross-reconstruction self-supervised pretraining method, which generate two vieww of original point cloud and train masked autoencoder to reconstruct one view by taking the inputs of another. The proposed method combine both contrastive learning and MAE learning into a single framework and demonstrate SOTA performance on few shot segmentation tasks.

**Strengths:**

1. This self-supervised training paradigm is quite intriguing. It combines contrastive learning with MAE-based completion to achieve better results.

2. The experiments conducted by the authors are highly convincing. The results in Table 4 demonstrate the benefits of decoupling views for representation learning, which can inspire future work.

**Weaknesses:**

1. In contrastive learning in image domain, the image augmentations typically preserve the main part of the image in different view, whereas in this setting,  the cropped point clouds often represent parts of an object. This will loses some contextual information. It's unclear why this limited information is sufficient to reconstruct the local part of point cloud from another view. This aspect needs further explanation.
2. There is some similar self-supervised approach has been proposed in the image domain, such as [1, 2]. Though they focus on different domain, the authors should consider discussing their difference in terms of learning framework in the related work section.

[1] Efficient Image Pre-Training with Siamese Cropped Masked Autoencoders
[2] Siamese Masked Autoencoders

**Questions:**

Why does this self-supervised paradigm significantly improve classification tasks (table 2), but offer limited improvements in segmentation tasks (table 3)?

---

> ### Author Response · Authors · 2024-11-20
> **Response to Q1-Q2**
>
> Thank you for taking the time to review our paper and for your valuable feedback. We greatly appreciate your recognition of the novelty and potential of our Point-PQAE. Below, we answer your questions in detail.
>
> > Q1. In contrastive learning in image domain, the image augmentations typically preserve the main part of the image in different view, whereas in this setting, the cropped point clouds often represent parts of an object. This will loses some contextual information. It's unclear why this limited information is sufficient to reconstruct the local part of point cloud from another view. This aspect needs further explanation.
>
> **First, there also exist contextual information loss in contrastive learning.** Although random crop in images typically preserves the main part of the image in different views, there are additional augmentations that can cause contextual information to be lost, such as color distortions and Gaussian blur, as used in the contrastive learning method SimCLR [5]. That is to say, models like SimCLR have demonstrated that utilizing limited contextual information for pre-training tasks is feasible.
>
> **Next, the reasons why our Point-PQAE can use limited information to reconstruct the local part** of the point cloud from another view can be mainly attributed to three factors:
>
> 1. **RPE provides sufficient guidance.** On one hand, RPE includes the geometry center difference between two views, which provides the relative position of the two views for the model to utilize. On the other hand, RPE includes the Positional Embeddings (patch centers) of target view for coarse reconstruction guidance.
> 2. We set the minimum crop ratio $r_m=0.6$, **ensuring that the main part of the point cloud is preserved in each view.** This makes the loss of some contextual information manageable.
> 3. The pre-training objective encourages the encoder not only to perceive inner-view information well, **but also to make the encoded representation implicitly contain global information (out-of-view information).** Then given the RPE, the reconstruction of the other view can then be inferred effectively.
>
>
> > Q2. There are some similar self-supervised approacha that have been proposed in the image domain, such as [1, 2]. Though they focus on different domain, The authors should consider discussing their difference in terms of learning framework in the related work section.
>
> Thank you for your valuable suggestion. **We have added a discussion of the two mentioned works in the Appendix A of the [revised PDF](https://openreview.net/pdf?id=jZASmAlxp2).** The discussion is as follows:
>
> There are two essential components in our proposed cross-reconstruction framework:
>
> 1. Two isolated/decoupled views, rather than two parts of the same instance that maintain a fixed relative relationship.
> 2. A model that achieves cross-view reconstruction using relative position information.
>
> **Our Point-PQAE pioneers the cross-reconstruction paradigm in 3D point cloud self-supervised learning (SSL)**, and to our knowledge, there are no similar previous methods in this domain. To better position our methodology, we compare it to similar SSL methods, including SiamMAE [1] and CropMAE [2], proposed in the image domain.
>
> **SiamMAE** operates on pairs of randomly sampled video frames and asymmetrically masks them, utilizing the past frame to predict the masked future frame. **CropMAE** relies on image augmentations to generate two views, using one to reconstruct the other.
>
> Relation to our Point-PQAE:
>
> 1. **All are cross-reconstruction methods.** Both SiamMAE and CropMAE have two essential components for cross-reconstruction-framework including multi-view (different frames sampled from one video for SiamMAE and isolated augmented images for CropMAE) and cross-view reconstruction model. They can be treated as cross-reconstruction methods, similar to our Point-PQAE.
>
> Difference:
>
> 1. **Different domain:** SiamMAE and CropMAE focus on the 2D SSL domain. Our Point-PQAE is the first method for cross-reconstruction in the 3D SSL domain.
> 2. **Asymmetric/symmetric reconstruction:** SiamMAE uses the past to predict the future, which is asymmetric. CropMAE performs asymmetric reconstruction and doesn't explore siamese cross-reconstruction. In contrast, our Point-PQAE is inherently symmetric, and the siamese loss brings a performance gain.
> 3. **No relative information utilized:** SiamMAE and CropMAE do not incorporate relative information into training but rely on non-fully masking to guide the cross-reconstruction. The RPE adopted by our Point-PQAE provides explicit guidance, making training more stable and improving explainability.
> 4. **No tuned-needed mask ratio exists in our framework.** There is a hyperparameter mask ratio that needs to be tuned in both SiamMAE and CropMAE, but this is not the case in our Point-PQAE framework.

---

> > ### Comment · Reviewer_eEi5 · 2024-11-22
> > **Lower my score due to the limited effectiveness**
> >
> > I have reservations about the effectiveness of using cross-view reconstruction, similar to the two reviewers (nNPL and Gtpy). Although the authors conducted numerous experiments, I did not observe the benefits of cross-view from the experimental results. I believe that during the process of using view1 to reconstruct view2, the information from view2 is already leaked to the features of view1 in the decoder's part. This approach does not essentially differ from reconstruction based on a single view. I acknowledge the concerns raised by nNPL and also noticed the limitations of this approach in their response to nNPL. Although the article is flawless in writing and illustrations, I believe that the feasibility and effectiveness of the idea are limited. Therefore, I have appropriately lowered my score.

---

> > > ### Author Response · Authors · 2024-11-23
> > > **Response Part 1**
> > >
> > > Thank you for your response and recognition of the presentation of our paper. We understand that there were some misunderstandings regarding the ablation experiment settings and the proposed techniques on the part of reviewer nNPL, which may have impacted your assessment of the effectiveness and novelty of our Point-PQAE. Now that most of these misunderstandings have been addressed, **we kindly ask if it would be possible to reconsider the ratings.**
> > >
> > > We would like to restate the misunderstood points briefly and address your concerns one by one:
> > > > Q1.1 Although the authors conducted numerous experiments, I did not observe the benefits of cross-view from the experimental results. (Effectiveness of cross-reconstruction paradigm)
> > >
> > > Regarding the effectiveness of the cross-view learning paradigm, we conducted ablations to compare it with its single-view counterpart.
> > >
> > > We define **two `parts` as components that maintain fixed relative relations.** These retain fixed relations between visible and masked parts in Point-MAE (or other self-reconstruction method such as line 1 in the table) because no independent decoupling operations or augmentations are applied to the different parts. In contrast, we define **two `views` as components that possess varying relative relations due to decoupling operations applied to the views.**
> > >
> > > The ablation experiment results are as follows:
> > >
> > > |Method|Multi-view|min-max norm part 1 (Decentralization)|min-max norm part 2 (Scale to [-1, 1])|Rotation|OBJ-BG|OBJ-ONLY|PB-T50-RS|
> > > -|-|-|-|-|-|-|-
> > > |Point-PQAE (APE)|✗|✗|✗|✗|92.3|91.0|87.7
> > > |Point-PQAE (APE)|✓|✗|✓|✗|94.1|92.1|88.2
> > > |Point-PQAE (APE)|✓|✗|✗|✓|93.6|92.4|88.8
> > > |Point-PQAE (APE)|✓|✗|✓|✓|93.8|92.8|88.8
> > > |Point-PQAE (RPE)|✓|✓|✓|✓|95.0|93.6|89.6
> > >
> > > **Recall:** After obtaining the cropped point cloud, **min-max normalization** is applied, similar to the process in 2D image tasks (e.g., recentering a cropped image and rescaling it to a fixed resolution, such as 224x224). This step isolates the coordinate systems between the parts, **ensuring the two parts are initially decoupled**. Subsequently, **the augmentation (Rotation) is introduced to further enhance variance** and achieve sufficiently decoupled views generation.
> > >
> > > **Min-max normalization part 1 (Decentralization)** makes APE infeasible. By using Point-PQAE (APE), we eliminate the compounding factor of RPE, necessitating the inclusion of ablation studies under the APE setting. Therefore, we decomposed min-max normalization into two components: Decentralization and Scaling to [-1, 1].
> > >
> > > The results clearly show that reconstructing one `part` from another `part` (Line 1) **leads to poor performance with Point-PQAE (APE), representing the single-view learning paradigm.** Conversely, when views are sufficiently decoupled, performance improves significantly (Lines 4/5 compared to Line 1 as the baseline). **This demonstrates the significant effectiveness of cross-`view` learning over cross-`part` (single-view learning).**
> > >
> > > **We acknowledge that there were previous misunderstandings about the comparison of our performance relative to baseline on the part of reviewer nNPL, which we have now addressed:**
> > >
> > > **Regarding the results:**
> > > - It's expected that **Point-PQAE (APE) Line 1** underperforms Point-MAE (Rotation) (accuracy: 92.6 91.9 88.4):
> > >     - Line 1 setting: crop two point clouds independently with $r_m=0.6$ without following view decoupling operations (min-max norm and rotation)
> > >     - As a result, the two point clouds overlap significantly and maintain fixed relative relations.
> > >     - This makes reconstructing one part from the other relatively trivial in the `single-view` setting—**even more trivial than in Point-MAE, which reconstructs non-overlapping parts.**
> > >     - Hence, Point-PQAE (APE) Line 1 underperforms Point-MAE (Rotation).
> > > - **It’s incorrect to directly compare each ablation result to Point-MAE.**
> > >     - Point-PQAE uses overlapping parts, whereas Point-MAE uses non-overlapping masked and visible parts.
> > >     - Similarly, in the 2D domain, CropMAE uses overlapping views while MAE uses non-overlapping masked and visible parts.
> > >     - **The appropriate comparison baseline is Point-PQAE (APE) Line 1, not Point-MAE.**
> > > - **It’s also incorrect to view Row 3 as the sole comparison between Point-PQAE and Point-MAE (Rotation).**
> > >     - Min-max normalization is an essential step for initial view decoupling, analogous to rescaling to 224x224 after cropping in the 2D domain.
> > >     - Please focus on the comparison of min-max normalization + rotation (Line 5) vs. Point-MAE (Rotation).

---

> ### Author Response · Authors · 2024-11-20
> **Response to Q3**
>
> > Q3. Why does this self-supervised paradigm significantly improve classification tasks (table 2), but offer limited improvements in segmentation tasks (table 3)?
>
> This is likely **because the fine-tuning model architecture differs between these two tasks**, which weakens the effect brought by different self-supervised learning methods (e.g., cross-reconstruction vs. self-reconstruction). Specifically:
>
> 1. **Segmentation requires fine-grained guidance**, but the use of average and max pooling during finetuning may not provide enough detail for effective segmentation, limiting pre-training benefits. In contrast, classification tasks require less fine-grained detail, so the pooled features are sufficient for good performance, fully demonstrating our pre-training advantage over the previous self-reconstruction method.
>
> 2. **Segmentation involves more layers trained from scratch.** In classification, only a simple classification head with three trainable MLPs is added to the pre-trained encoder. However, in segmentation, six additional Conv1d layers are introduced, some with large dimensions (e.g., 1153 input channels and 1536 output channels), increasing the model's complexity and potentially hindering the effectiveness of pre-training.
>
> 3. **Segmentation involves a more complex dataflow.** In classification, the dataflow is straightforward: after encoding the input point cloud, the latent representation is passed directly into the classification head, **which directly shows the effectiveness of different pre-training methods.** In contrast, segmentation uses outputs from multiple layers of the pre-trained encoder (4th, 8th, and 12th layers), and introduces new layers to process the raw point cloud data. This is then concatenated with the global features for the final segmentation, adding complexity to the model's dataflow.
>
> Considering the above reasons, our cross-reconstruction Point-PQAE improves classification tasks significantly but offers limited improvements in segmentation tasks compared to the self-reconstruction baseline, Point-MAE. Other peer methods also show similar trends, e.g., ACT [3] and ReCon [4].
>
>
>
> Reference
>
>
> [1] Gupta A, Wu J, Deng J, et al. Siamese masked autoencoders[J]. NeurIPS 2023.
>
> [2] Eymaël A, Vandeghen R, Cioppa A, et al. Efficient Image Pre-Training with Siamese Cropped Masked Autoencoders[C]//ECCV 2024.
>
> [3] Dong R, Qi Z, Zhang L, et al. Autoencoders as Cross-Modal Teachers: Can Pretrained 2D Image Transformers Help 3D Representation Learning?[C]//ICLR 2023.
>
> [4] Qi Z, Dong R, Fan G, et al. Contrast with reconstruct: Contrastive 3d representation learning guided by generative pretraining[C]//ICML 2023.
>
> [5] Chen T, Kornblith S, Norouzi M, et al. A simple framework for contrastive learning of visual representations[C]//ICML 2020.

---

> ### Author Response · Authors · 2024-11-23
> **Response Part 2**
>
> > Q1.2 I believe that during the process of using view1 to reconstruct view2, the information from view2 is already leaked to the features of view1 in the decoder's part.
>
> Thank you for your good question. **It’s worth noting that appropriate information leakage or provision is essential for many self-supervised learning (SSL) methods:**
>
> - **In the image domain**, to reconstruct masked parts, **visible parts are provided to the decoder in MAE [1]**. Similarly, to reconstruct view 2, the encoded **non-fully masked view 2 is provided (or “leaked”)** to the decoder in methods such as SiamMAE [2] and CropMAE [3], enabling cross-attention with the encoded fully-visible view 1. Without this provision of non-fully masked view 2, cross-reconstruction would not be achievable.
>
> - **In the 3D point cloud domain** (e.g., Point-MAE), to reconstruct masked parts, the decoder not only receives visible points but also **positional embeddings (centers) of the masked patches**. Without these leaked positional embeddings, reconstruction becomes infeasible. This is because predicted points are decentralized based on their corresponding centers, **so accurate masked centers are necessary as coarse shape guidance.**
>
> In our method, the interaction between the two views occurs only through the RPE, which is processed by the Positional Query module. Specifically, the RPE for reconstructing view 1 contains two key components:
>
> 1. **The view geometry center difference information** for modeling relative view positions.
> 2. **The positional embeddings (centers) of view 1**, which provide coarse shape guidance similar to Point-MAE. **This guidance is essential for prediction of decentralization points and handling dynamic relative relations (inferring the isolatedly performed augmentations)**, and without positional embeddings, cross-reconstruction would not be possible.
>
> For these reasons, we view the information "leakage" from view 1 in the RPE as not only acceptable but also necessary for effective cross-reconstruction.
>
> > Q1.3 This approach does not essentially differ from reconstruction based on a single view. I acknowledge the concerns raised by nNPL and also noticed the limitations of this approach in their response to nNPL.
>
> **We have clarified the differences between our cross-reconstruction paradigm and single-view reconstruction in the response to Q1.1**, which can be summarized as follows:
>
> In single-view reconstruction, parts maintain fixed relative relations, whereas in our paradigm, decoupled views exhibit dynamic relative relations. The Point-PQAE pipeline is specifically designed to cross-reconstruct these decoupled views. **The table provided in Q1.1 highlights the effectiveness of our cross-view paradigm compared to the single-view paradigm.**
>
> Additionally, the concerns raised by nNPL regarding the effectiveness and limitations of our approach have already been well addressed. Please refer to the response to Q1.1 and our detailed reply to nNPL.
>
> Reference
>
> [1] He K, Chen X, Xie S, et al. Masked autoencoders are scalable vision learners[C]//CVPR 2022.
>
> [2] Gupta A, Wu J, Deng J, et al. Siamese masked autoencoders[J]. NeurIPS 2023.
>
> [3] Eymaël A, Vandeghen R, Cioppa A, et al. Efficient Image Pre-Training with Siamese Cropped Masked Autoencoders[C]//ECCV 2024.
>
> ### **Final Words:** We would greatly appreciate and highly value your response and reconsideration.

---

### Official Review · Reviewer_Gtpy · 2024-11-08

**Soundness:** 2
**Presentation:** 3
**Contribution:** 2
**Rating:** 5
**Confidence:** 4

**Summary:**

The authors recognize and explore a two-view pre-training paradigm and propose PointPQAE, a cross-reconstruction generative framework. PointPQAE first generates two decoupled point clouds/views and then reconstructs one from the other. Additionally, PointPQAE achieves new state-of-the-art results, surpassing previous single-modal self-reconstruction methods in 3D self-supervised learning.

**Strengths:**

1. The writing is clear and easy to understand.
2. The authors conducted numerous experiments to verify the feasibility of the method.

**Weaknesses:**

1. In some metrics, the method performs similarly to PointM2AE, so I find the claim that it "surpasses previous single-modal self-reconstruction methods in 3D self-supervised learning by a large margin" to be overly bold.

2. After reading the introduction, I am still somewhat confused as to why this design works: (1) Why is Part 1 able to recover its own structural information through the structural information of Part 2? What if Part 1 and Part 2 do not overlap at all? (2) What is the approximate overlap ratio between Part 1 and Part 2? Could you provide statistical data on this?

3. Limited novelty: The core idea of cross-reconstruction has already been evident in numerous previous works [1][2][3], making it difficult for me to identify any particularly compelling innovation in this study.

Reference:
[1] Guo, Ziyu, et al. "Joint-mae: 2d-3d joint masked autoencoders for 3d point cloud pre-training." arXiv preprint arXiv:2302.14007 (2023).
[2] Chen, Anthony, et al. "Pimae: Point cloud and image interactive masked autoencoders for 3d object detection." Proceedings of the IEEE/CVF Conference on Computer Vision and Pattern Recognition. 2023.
[3] Gupta, Agrim, et al. "Siamese masked autoencoders." Advances in Neural Information Processing Systems 36 (2023): 40676-40693.

**Questions:**

See the weaknesses.

---

> ### Author Response · Authors · 2024-11-20
> **Response to Q1-Q2**
>
> Thank you for the time, thorough comments, and nice suggestions. We are pleased to clarify your questions one-by-one.
>
> > Q1. In some metrics, the method performs similarly to Point-M2AE, so I find the claim that it "surpasses previous single-modal self-reconstruction methods in 3D self-supervised learning by a large margin" to be overly bold.
>
> Thank you for your suggestion. We have changed this expression into "surpasses previous single-modal self-reconstruction methods in 3D self-supervised learning by a ~~large~~ margin". Please refer to line 27 in our uploaded [revised PDF](https://openreview.net/pdf?id=jZASmAlxp2).
>
> However, **it's important to note that Point-M2AE adopts a hierachical transformer architecture and advanced Multi-scale Masking strategy while our Point-PQAE adopts vanilla architecture and strategy.** And our Point-PQAE outperforms the most relavant baseline (Point-MAE) which also adopts vanilla transformer architecture and strategy by a large margin in most metrics.
>
> As illustrated in Appendix E, compared to the well-studied self-reconstruction, cross-reconstruction remains significantly under-explored. Given that vanilla architectures may not be ideal for cross-reconstruction, we believe that advanced (e.g., hierarchical) architectures or specialized cross-reconstruction strategies may further enhance performance, which we leave for future work.
>
> > Q2.1 Why is Part 1 able to recover its own structural information through the structural information of Part 2? What if Part 1 and Part 2 do not overlap at all?
>
> The reasons include that the **Relative Positional Embedding (RPE) provides sufficient guidance for cross-reconstruction** and **the pre-training objective encourages the model to learn both the inter-positional relationships between two views and the inner spatial information within the views.**
>
> To recover view 2, $RPE_{1→2}$ (see Eq. 6 and Eq. 7) is calculated. $RPE_{1→2}$ includes the geometric center difference between the two views for relative guidance and incorporates the Positional Embeddings (patch centers) within view 2 for coarse shape guidance. This provides enough information for the model to infer the independently augmented view 2 using the known view 1.
>
> The pre-training objective requires the model to infer one view from the other. If the encoder only encodes view 2 without capturing implicit global shape information, cross-reconstruction would not be possible, even with sufficient RPE. In other words, the pre-training encourages the encoder to capture both local (within-view) and global (out-of-view) information. Then with the RPE, reconstruction can then be achieved.
>
> **Thus, whether the views overlap or not is not critical, because sufficient information is provided.** As shown in Figure 3 (crop ratio 20%), the body of the plane is used to recover the tail of the plane. These are non-overlapping views, but the reconstruction still works well. However, low overlap ratios introduce greater difficulty for pre-training, which is why we choose a minimum crop ratio of $r_m = 0.6$ (see the ablation study in Figure 4).
>
> > Q2.2 What is the approximate overlap ratio between Part 1 and Part 2? Could you provide statistical data on this?
>
> We conducted experiments with different values of the minimum crop ratio $r_m$ (line 191 in the paper) and report the approximate overlap ratio between the two views across the entire point cloud, as shown below:
>
> $r_m$|0.2|0.3|0.4|0.5|**0.6**|0.7|
> -|-|-|-|-|-|-
> Overlap ratio|0.3677|0.4459|0.5146|0.5937|**0.6713**|0.7501
>
> We adopted $r_m = 0.6$, as it performs the best in practice (see the ablation in Fig. 4). Choosing an appropriate $r_m$ is crucial: if it is too high, excessive overlap can simplify the cross-reconstruction task, while if it is too low, the task becomes overly difficult, hindering the learning of semantic representations.

---

> ### Author Response · Authors · 2024-11-20
> **Response to Q3**
>
> > Q3. Limited novelty: The core idea of cross-reconstruction has already been evident in numerous previous works [1][2][3], making it difficult for me to identify any particularly compelling innovation in this study.
>
> Thank you for reminding us to discuss related works to highlight the novelty of our method. **It is important to emphasize that our proposed Point-PQAE differs from these three methods both in terms of motivation and framework.**
>
> There are two essential components in our proposed cross-reconstruction framework:
>
> 1. **Two isolated/decoupled views, rather than two parts** of the same instance that maintain a fixed relative relationship.
> 2. A model that achieves cross-view reconstruction using relative position information.
>
> **First, we discuss the closely related method SiamMAE [1], which is also mentioned by reviewer eEi5.**
>
> The relation of SiamMAE to our Point-PQAE:
>
> 1. **Both are cross-reconstruction methods.** SiamMAE has two essential components: multi-view (different frames sampled from one video) and a cross-view reconstruction model, and can be treated as a cross-reconstruction method.
>
> Difference:
>
> 1. **Different domain:** SiamMAE focus on the 2D SSL domain. Our Point-PQAE is the first method for cross-reconstruction in the 3D SSL domain.
> 2. **Asymmetric/symmetric reconstruction:** SiamMAE uses the past to predict the future, which is asymmetric. In contrast, our Point-PQAE is inherently symmetric, and the siamese loss brings a performance gain.
> 3. **No relative information utilized:** SiamMAE does not incorporate relative information into training but rely on non-fully masking to guide the cross-reconstruction between past and future frames. The RPE adopted by our Point-PQAE provides explicit guidance, making training more stable and improving explainability.
> 5. **No tuned-needed mask ratio exists in our framework.** There is a hyperparameter mask ratio that needs to be tuned in both SiamMAE, but this is not the case in our Point-PQAE framework.
>
> Then, we discuss the differences between our framework and the other two methods in the same domain, including **Joint-MAE [2] and PiMAE [3]**. Joint-MAE and PiMAE adopt a similar strategy, utilizing paired point clouds and images to perform masked autoencoding. **Our framework differs significantly from these two methods.**
>
> The relation of them to our Point-PQAE:
>
> 1. These three methods are all self-supervised methods that focus on the point cloud domain.
>
> Difference:
>
> 1. **Different motivations:** Joint-MAE and PiMAE aim to explore the semantic correlation between 2D and 3D by performing 3D-2D interactions and achieving cross-modal self-reconstruction through cross-modal knowledge. In contrast, drawing inspiration from the success of two-view pre-training paradigms, we propose Point-PQAE, the first cross-reconstruction framework for point cloud self-supervised learning (SSL).
>
> 2. **Different modality:** Both Joint-MAE and PiMAE rely on **paired image-point cloud data**, making them cross-modal methods. Our Point-PQAE only consumes unlabeled **point cloud data**, which can be more easily extended. Additionally, incorporating image data could increase computational requirements.
>
> 3. **Joint-MAE and PiMAE cannot be strictly called cross-reconstruction methods because:**
>     - Recall that the two components for cross-reconstruction methods are decoupled views and a cross-reconstruction framework. In cross-reconstruction, **decoupled views are obtained through independent augmentations, achieving significant diversity between views, and the cross-reconstruction framework relies on information from view 1 to mandatorily reconstruct view 2.**
>     - **The paired 3D and 2D views used by Joint-MAE and PiMAE cannot be considered isolated/decoupled views.** Take PiMAE as an example: The image is merely a render from a specific camera pose of the point cloud. No augmentations can be applied to either of these views (as discussed in Section 4 of the PiMAE paper), so diversity between views cannot be achieved.
>     - **Cross-view knowledge is used as auxiliary, not mandatory, in these two methods.** If either the 3D or 2D data is removed, reconstruction can still be achieved, which is precisely the case in MAE [4] or Point-MAE [5]. However, a cross-reconstruction framework should mandatorily relies on view 1 to reconstruct view 2, including in our Point-PQAE, SiamMAE [1], and CropMAE [6]. For instance, in Joint-MAE, 3D information is used as auxiliary for 2D MAE (or vice versa), and a cross-reconstruction loss (specifically, cross-modal reconstruction loss) is added to the 2D-3D output.
>
>     **Thus, it is more appropriate to call them cross-modality self-reconstruction methods.**

---

> ### Author Response · Authors · 2024-11-20
> **Reference**
>
> [1] Gupta A, Wu J, Deng J, et al. Siamese masked autoencoders[J]. NeurIPS 2023.
>
> [2] Guo Z, Zhang R, Qiu L, et al. Joint-MAE: 2D-3D joint masked autoencoders for 3D point cloud pre-training[C]//IJCAI 2023.
>
> [3] Chen A, Zhang K, Zhang R, et al. Pimae: Point cloud and image interactive masked autoencoders for 3d object detection[C]//CVPR 2023.
>
> [4] He K, Chen X, Xie S, et al. Masked autoencoders are scalable vision learners[C]//CVPR 2022.
>
> [5] Pang Y, Wang W, Tay F E H, et al. Masked autoencoders for point cloud self-supervised learning[C]//ECCV 2022.
>
> [6] Eymaël A, Vandeghen R, Cioppa A, et al. Efficient Image Pre-Training with Siamese Cropped Masked Autoencoders[C]//ECCV 2024.

---

> ### Author Response · Authors · 2024-11-24
> **Look forward to your further reply**
>
> Dear Reviewer Gtpy,
>
> Thank you again for your valuable feedback and constructive suggestions.
>
> We have revised the PDF, illustrated the principle of cross-reconstruction and compared our Point-PQAE with the mentioned methods to highlight the novelty of our work.
>
> As the discussion phase is coming to an end, **we would like to know if our rebuttal has addressed your concerns.** If you have any further questions or need additional clarification, please feel free to let us know.

---

> ### Comment · Reviewer_Gtpy · 2024-11-25
>
> I sincerely thank the author for their response and for addressing my concerns. I fully agree with the issues and limitations raised by the other reviewers, particularly nNPL, regarding the proposed methods. Therefore, I have decided to maintain my score.

---

### Author Response · Authors · 2024-11-27
**Global official comment**

We would like to summarize the rebuttal (addressed concerns and inspiring explorations) for your convenience in reviewing the overall conditions:

- We compare our Point-PQAE with SiamMAE, CropMAE, Joint-MAE, and PiMAE to **highlight the great novelty** of our Point-PQAE (to Reviewer Gtpy and eEi5).
- We repeatedly illustrate the RPE technique to Reviewer nNPL, ultimately addressing his/her misunderstanding, **clarifying the distinction between our RPE and existing RPEs** (to Reviewer nNPL).
- We demonstrate the **effectiveness of our cross-view learning paradigm** with ablations using Point-PQAE (APE) and Point-PQAE (RPE) (to Reviewer nNPL):
    - The setting for this ablation was initially misunderstood by Reviewer nNPL. After clarification, he/she agrees with our effectiveness and performance gain.
    - Unfortunately, Reviewer eEi5 followed the misunderstanding, lowered their scores, and did not respond after this concern was addressed.
    - Reviewer Gtpy also followed the same concern, so we are trying to confirm whether this issue related to effectiveness is well addressed and encourage Reviewer eEi5 and Gtpy to reconsider it.
    - **The ablation summary**: (See line 3 vs. line 1; with sufficient view decoupling and our proposed technique, cross-view reconstruction significantly outperforms the single-view case, **demonstrating the great effectiveness of our proposed cross-reconstruction framework**.)


    |Method|Multi-view|min-max norm part 1 (Decentralization)|min-max norm part 2 (Scale to [-1, 1])|Rotation|OBJ-BG|OBJ-ONLY|PB-T50-RS|
    -|-|-|-|-|-|-|-
    |Point-PQAE (APE)|✗|✗|✗|✗|92.3|91.0|87.7
    |Point-PQAE (APE)|✓|✗|✓|✓|93.8|92.8|88.8
    |Point-PQAE (RPE)|✓|✓|✓|✓|95.0|93.6|89.6

- We discuss the potential of cross-reconstruction being applicated to other domains including text and image (to Reviewer qqpL).
- We explore the **triple-reconstruction setting** and find that it is a promising pre-training acceleration technique that can achieve competitive performance with less training time than double-reconstruction (to Reviewer qqpL).

---

### Meta-Review · Area_Chair_LNfF · 2024-12-16

**Metareview:**

This paper receives 3 negative ratings and 2 positive ratings. Although the paper has some merits, e.g., thorough experiments, the reviewers pointed out a few critical concerns about 1) technical novelty and soundness of using cross-view reconstruction, 2) effectiveness of the proposed method, 3) experimental results. After taking a close look at the paper, rebuttal, and discussions, the AC agrees with reviewers' feedback and hence suggests the rejection decision. The authors are encouraged to improve the paper based on the feedback for the next venue.

**Additional Comments On Reviewer Discussion:**

In the rebuttal, some of the concerns like technical clarity are addressed by the authors. However, during the post-rebuttal discussion period, the reviewer nNPL is not convinced about a few limitations of the work, e.g., baseline choices, minor improvements over other methods, multi-view learning in the proposed approach. Despite that the discussed thread is very active where the authors also provided many explanations and additional results, the reviewer is still concerned about these limitations. In addition, the reviewer Gtpy, eEi5, and qqpL (who intended to downgrade the rating but not allowed in the system) also agree with reviewer nNPL's feedback, while the remaining reviewer d4im who provided the positive rating only has a very short review and did not participate in the discussion. The AC hence took a close look at the discussion thread and agrees with the reviewers that these issues should be significantly improved in the manuscript, which still requires a good amount of effort to make the paper ready for publication.

---

### Decision · Program_Chairs · 2025-01-22

Reject